   

# Divergent roles of RIPK3 and MLKL in high-fat diet–induced obesity and MAFLD in mice

Hazel Tye[1,*], Stephanie A Conos[1,2,*], Tirta M Djajawi[1,2,*], Timothy A Gottschalk[1,2,*], Nasteho Abdoulkader[1,†], Isabella Y Kong[3,4,†], Helene L Kammoun[5], Vinod K Narayana[6], Tobias Kratina[7], Mary Speir[1,2], Jack Emery[1,2], Daniel S Simpson[3,4], Cathrine Hall[3], Angelina J Vince[3], Sophia Russo[3], Rhiannan Crawley[3], Maryam Rashidi[3,4], Joanne M Hildebrand[3,4], James M Murphy[3,4,8], Lachlan Whitehead[3,4], David P De Souza[6], Seth L Masters[3,4], Andre L Samson[3,4], Najoua Lalaoui[3,7,9], Edwin D Hawkins[3,4], Andrew J Murphy[5], James E Vince[3,4], Kate E Lawlor[1,2,3,4]

**Cell death frequently occurs in the pathogenesis of obesity and metabolic dysfunction–associated fatty liver disease (MAFLD). However, the exact contribution of core cell death machinery to disease manifestations remains ill-defined. Here, we show via the direct comparison of mice genetically deficient in the essential necroptotic regulators, receptor-interacting protein kinase-3 (RIPK3) and mixed lineage kinase domain–like (MLKL), as well as mice lacking apoptotic caspase-8 in myeloid cells combined with RIPK3 loss, that RIPK3/caspase-8 signaling regulates macrophage inflammatory responses and drives adipose tissue inflammation and MAFLD upon high-fat diet feeding. In contrast, MLKL, divergent to RIPK3, contributes to both obesity and MAFLD in a manner largely independent of inflammation. We also uncover that MLKL regulates the expression of molecules involved in lipid uptake, transport, and metabolism, and congruent with this, we discover a shift in the hepatic lipidome upon MLKL deletion. Collectively, these findings highlight MLKL as an attractive therapeutic target to combat the growing obesity pandemic and metabolic disease.**

## Introduction

Obesity is a global pandemic associated with the consumption of a Western diet rich in saturated fats and refined carbohydrate, and a sedentary lifestyle. Diet-induced obesity leads to increased circulating free fatty acids, insulin, and glucose that trigger "low-grade" inflammation that is causally related to inflammatory macrophage expansion within the hypertrophic adipose tissue (Christ & Latz, 2019). This state of chronic metabolic inflammation predisposes obese individuals to multi-organ insulin resistance and comorbidities, such as metabolic dysfunction–associated fatty liver disease (MAFLD). Evidence implicates gut dysbiosis and a "leaky gut" in systemic inflammation via the release of damaging microbial products (e.g., lipopolysaccharide) (Hersoug et al, 2016) that act synergistically with excess dietary metabolites, such as saturated fatty acids, to potentiate inflammatory signaling and cytokine/chemokine production (Lancaster et al, 2018). Obesity-induced inflammation is also inextricably linked to the activation and assembly of the NOD-like receptor protein 3 (NLRP3) inflammasome through the sensing of various metabolic DAMPs (e.g., palmitic acid, cholesterol crystals) (Liang et al, 2021). NLPR3 inflammasome–associated caspase-1 activity subsequently cleaves and activates IL-1$\beta$, as well as the pyroptotic effector gasdermin D (GSDMD) (Camell et al, 2015; Yabal et al, 2019). Intriguingly, despite clear evidence that NLRP3 inflammasome and IL-1$\beta$ activity drive obesity and MAFLD (Stienstra et al, 2010, 2011; Vandanmagsar et al, 2011; Coll et al, 2022), whether pyroptosis or other modes of programmed cell death, such as apoptosis and necroptosis, facilitate the demise of key cell types in tissues remains ambiguous.

Targeting the extensive hepatocyte death observed in MAFLD progression has emerged as an attractive therapeutic option to prevent subsequent liver pathologies. Saturated fatty acids are thought to dominantly induce apoptosis in hepatocytes via extrinsic death receptor signaling, and through ER stress, oxidative stress, dysregulated autophagy, and c-Jun N-terminal kinase (JNK) 1 signaling, which all culminate in mitochondrial dysfunction (Akazawa & Nakao, 2018). In the case of extrinsic apoptosis, formation of a death-inducing signaling complex, consisting of receptor-

[1]Centre for Innate Immunity and Infectious Diseases, Hudson Institute of Medical Research, Clayton, Australia   [2]Department of Molecular and Translational Science, Monash University, Clayton, Australia   [3]The Walter and Eliza Hall Institute of Medical Research, Parkville, Australia   [4]The Department of Medical Biology, University of Melbourne, Parkville, Australia   [5]Baker Heart and Diabetes Institute, Melbourne, Australia   [6]Metabolomics Australia, Bio21 Institute of Molecular Science and Biotechnology, University of Melbourne, Melbourne, Australia   [7]Peter MacCallum Cancer Centre, Melbourne, Australia   [8]Drug Discovery Biology, Monash Institute of Pharmaceutical Sciences, Monash University, Parkville, Australia   [9]Sir Peter MacCallum Department of Oncology, University of Melbourne, Melbourne, Australia

Correspondence: kate.lawlor@hudson.org.au
*Hazel Tye, Stephanie A Conos, Tirta M Djajawi, and Timothy A Gottschalk contributed equally to this work
†Nasteho Abdoulkader and Isabella Y Kong contributed equally to this work

interacting protein kinase-1 (RIPK1) and/or FADD/caspase-8, may be triggered upon death receptor (DR5, TNFR) or TLR4 ligation, when pro-survival signals are compromised (e.g., upon IAP loss) (Feltham et al, 2017). Fitting with a key role of apoptotic caspase-8 in MAFLD, deletion of caspase-8 in hepatocytes protected mice from methionine- and choline-deficient (MCD) diet–induced liver injury and inflammation (Hatting et al, 2013), and pan-caspase inhibitor emricasan attenuated liver injury, inflammation, and fibrosis from high-fat diet (HFD) intake (Barreyro et al, 2015). Conflictingly, caspase-8 deletion in mouse liver parenchymal cells exacerbated MCD diet–induced liver damage (Gautheron et al, 2014), and emricasan underperformed in phase II clinical trials and sometimes worsened progression of metabolic dysfunction–associated steatohepatitis (MASH) (Lekakis & Cholongitas, 2022), suggesting that a form of caspase-independent cell death is triggered.

Necroptosis is a lytic form of cell death that is unleashed in the absence of caspase-8 activity and is critically dependent on RIPK1/3 kinase activity and the pseudokinase mixed lineage kinase domain–like (MLKL). Formation of the necrosome triggers RIPK3-mediated phosphorylation of MLKL, prompting a conformational change that allows the N-terminal four-helical bundle domain to associate with membranes, causing cell lysis (Murphy et al, 2013; Hildebrand et al, 2014; Wang et al, 2014; Samson et al, 2020; Garnish et al, 2021). Although the role of upstream kinases RIPK1 and RIPK3 in obesity and MAFLD remain debatable (Afonso et al, 2015, 2021; Gautheron et al, 2016; Roychowdhury et al, 2016; Karunakaran et al, 2020; Majdi et al, 2020; Tao et al, 2021), there are numerous reports that MLKL deficiency protects mice from MAFLD (Saeed et al, 2019; Xu et al, 2019; Wu et al, 2020). As RIPK3 can regulate the death- and inflammation-inducing activity of both caspase-8 and MLKL in disease-causing macrophages (Vince et al, 2012; Lawlor et al, 2015), direct comparisons of mutant animals in the same obesity and MAFLD model are required to define their pathological roles and divergent activities.

Significant plasticity has been shown between apoptotic, pyroptotic, and necroptotic cell death signaling pathways (Newton et al, 2019; Doerflinger et al, 2020; Karki et al, 2021; Simpson et al, 2022; Hughes et al, 2023). Moreover, crosstalk between intrinsic apoptosis, extrinsic apoptosis, and necroptosis with NLRP3 inflammasome and IL-1β activation has been documented in innate immune cells (Kang et al, 2013; Yabal et al, 2014; Lawlor et al, 2015; Wicki et al, 2016; Conos et al, 2017; Chen et al, 2018; Vince et al, 2018; Speir et al, 2023) and in vivo in models of inflammatory disease and infection (Rickard et al, 2014a; Lawlor et al, 2015; Gurung et al, 2016; Shouval et al, 2016; Polykratis et al, 2019; Deo et al, 2020; Speir & Lawlor, 2021). As both RIPK1(RIPK3)/caspase-8 and MLKL signaling can be triggered by dietary metabolite excess, it remains to be seen whether either of these cell death modes act upstream of NLRP3, or act in parallel, to drive distinct aspects of pathology. These studies will be vital to the advancement of therapies in the area and for predicting disease outcomes. Here, we directly contrast the contribution of RIPK3, along with caspase-8 in myeloid cells, and MLKL signaling in obesity and MAFLD development. Our study reveals that caspase-8 induces damaging inflammation to saturated fatty acids, whereas MLKL uniquely regulates obesity and MAFLD via non-canonical actions on lipid metabolism.

# Results

## Caspase-8 contributes to inflammasome priming, IL-1β activation, and cell death upon LPS and palmitate exposure

The NLRP3 inflammasome is well documented to activate IL-1β in macrophages in response to saturated fatty acid palmitate (Wen et al, 2011). Nevertheless, how NLRP3 is triggered by metabolic stress remains controversial (Yabal et al, 2019). As it has recently become apparent that both extrinsic apoptotic RIPK1/3/caspase-8 and necroptotic RIPK3/MLKL signaling can culminate in potassium ion efflux–dependent NLRP3 inflammasome activation (Conos et al, 2017), we queried whether caspase-8 or MLKL activity could be the trigger for NLRP3 inflammasome activity upon dietary stress. As expected, our analysis of NLRP3-deficient ($Nlrp3^{-/-}$) and caspase-1–deficient ($Casp1^{-/-}$) bone marrow-derived macrophages (BMDMs) confirmed that IL-1β activation in response to increasing concentrations of palmitate conjugated to BSA (PA-BSA) is largely NLRP3 inflammasome–dependent (Fig S1A and B), with residual IL-1β p17 activity most likely attributable to caspase-8–mediated IL-1β cleavage (Fig 1A) (Maelfait et al, 2008; Vince et al, 2012). In comparison, TNF secretion (a marker of inflammasome priming) was amplified in both WT and inflammasome-deficient BMDMs after LPS and palmitate treatment (Fig S1C). Unexpectedly, examination of cell death responses at 18–20 h post-LPS and palmitate exposure revealed that cell death was not perturbed in $Nlrp3^{-/-}$ and $Casp1^{-/-}$ BMDMs (Fig S1D), which contrasts the complete block in both IL-1β activation and pyroptotic cell death observed in the absence of NLRP3 or caspase-1 in response to canonical NLRP3 inflammasome stimulus nigericin (Fig S1E–G). These results suggest that although palmitate induces NLRP3 inflammasome activation in LPS-primed macrophages, ultimately, cellular demise occurs independent of inflammasome-associated pyroptosis.

To ascertain whether extrinsic apoptotic and necroptotic signaling regulates NLRP3 inflammasome activation and/or cell death upon macrophage exposure to palmitate, we next analyzed responses in BMDMs lacking the terminal necroptotic effector MLKL, or the essential kinase for necrosome formation and ripoptosome/complex II scaffold component RIPK3, or both RIPK3 and apoptotic caspase-8. It is worth noting that caspase-8 deficiency alone induces lethal necroptotic signaling during embryogenesis, which can be rescued by RIPK3 loss (Kaiser et al, 2011; Oberst et al, 2011). Interestingly, IL-1β activation was completely blocked in LPS-primed $Ripk3^{-/-}Casp8^{-/-}$ BMDMs after 18–20 h of palmitate (PA-BSA) stimulation, whereas loss of MLKL or RIPK3 had no impact (Figs 1A and B and S2A), suggesting that caspase-8 activity (marked by p43 and p18 cleavage products) regulates NLRP3 inflammasome and IL-1β activity. However, in line with previous reports of a transcriptional role of caspase-8 in inflammasome priming (Allam et al, 2014; Gurung et al, 2014), LPS-primed $Ripk3^{-/-}Casp8^{-/-}$ BMDMs exhibited reduced pro-IL-1β, NLRP3, and TNF levels compared with WT cells (Fig 1A and C) and, accordingly, blunted inflammasome responses to nigericin (Fig S2B and C). Importantly, the presence of caspase-1 p20 (marker of activity) in LPS- and PA-BSA–treated $Ripk3^{-/-}Casp8^{-/-}$ BMDMs (Fig 1A), albeit reduced compared with WT, indicates that caspase-8 is not essential for palmitate-induced NLRP3 activation. Consistent with this, after TLR2 priming with Pam3Cys, which

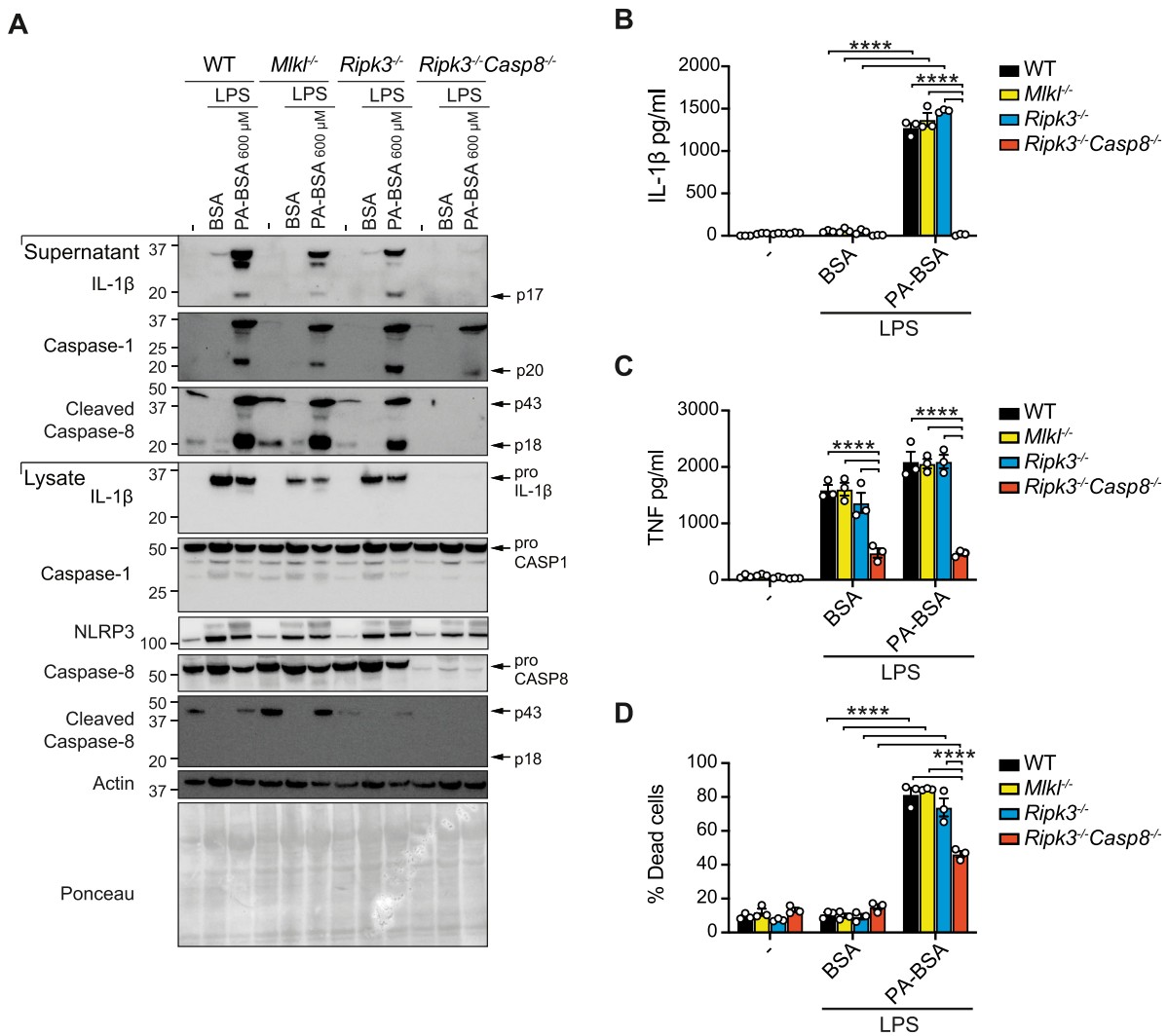

**Figure 1. NLRP3 inflammasome priming, IL-1β activation, and cell death are defective in *Ripk3⁻/⁻Casp8⁻/⁻* macrophages in response to LPS and palmitate.**
**(A, B, C, D)** WT, *Mlkl⁻/⁻*, *Ripk3⁻/⁻* and *Ripk3⁻/⁻Casp8⁻/⁻* BMDMs were primed with or without LPS (50 ng/ml) for 3 h and treated with 600 μM PA-BSA or BSA alone, as indicated, for 18–20 h. **(A)** Cell lysates and supernatants were analyzed by immunoblot for specified proteins. The results shown are representative of two independent biological experiments. **(B)** IL-1β and (C) TNF levels were measured in cell supernatants by ELISA. Data shown are the mean ± SEM of n = 3 biological replicates and are representative of at least four independent experiments. One-way ANOVA followed by Tukey's multiple comparison test, ****$P$ < 0.0001. **(D)** Cell viability was assessed by PI uptake and flow cytometric analysis. Data shown are the mean ± SEM of n = 3 biological replicates and are representative of at least four independent experiments. One-way ANOVA followed by Tukey's multiple comparison test, ****$P$ < 0.0001.
Source data are available for this figure.

is less dependent on caspase-8–induced transcriptional (Allam et al, 2014) and post-translational effects (Kang et al, 2015), we observed that palmitate triggered normal caspase-1 activity and low levels of mature IL-1β p17 secretion in *Ripk3⁻/⁻Casp8⁻/⁻* BMDMs that were associated with better priming responses (Fig S2D–F). Parallel analysis of cell death responses revealed that RIPK3 and MLKL were also not required for TLR- and PA-BSA–induced cell death at 20 h, whereas caspase-8 deficiency modestly reduced cell death in LPS-primed cells (Figs 1D and S2G). Collectively, these results suggest that neither MLKL nor caspase-8 lies upstream of NLRP3. Rather, caspase-8 regulates inflammasome priming in macrophages to LPS, in addition to partly contributing to IL-1β proteolysis and cell death upon palmitate exposure.

### RIPK3/caspase-8 signaling in myeloid cells contributes to obesity-induced metabolic dysfunction

Based on our observation that caspase-8 signaling regulated cell death and inflammatory responses in macrophages to LPS and palmitate in vitro, as well as past reports that RIPK3 and/or caspase-8 activity can regulate inflammasome-dependent and independent inflammatory responses in other disease models (Allam et al, 2014; Lawlor et al, 2015), we next investigated how loss of caspase-8 in myeloid cells impacts HFD-induced obesity and MAFLD development. *Ripk3⁻/⁻Casp8⁻/⁻* mice develop an autoimmune lymphoproliferative syndrome that precludes their long-term analysis in HFD obesity models (Kaiser et al, 2011; Oberst

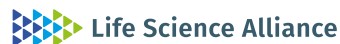

et al, 2011). Similarly, conditional, myeloid-specific deletion of caspase-8 using the lysozyme M-Cre transgene can lead to the deletion of myeloid progenitor cells by necroptosis when efficiently deleted, and in surviving cells, caspase-8 is often incompletely deleted (Kang et al, 2004), thereby reducing the utility of the $Casp8^{LysMcre}$ mouse. Hence, we chose to compare $Ripk3^{-/-}$ mice with mice lacking caspase-8 conditionally in myeloid cells on a RIPK3-deficient background ($Casp8^{LysMcre}Ripk3^{-/-}$) (Vince et al, 2018) that will rescue this loss of progenitors and ensure a more robust deletion of caspase-8 in myeloid cells. On a normal chow diet (ND), $Casp8^{LysMcre}Ripk3^{-/-}$ and $Casp8^{lox/lox}Ripk3^{-/-}$ (RIPK3-deficient) mice displayed no major difference in weights with aging compared with $Casp8^{lox/lox}Ripk3^{+/+}$ control mice (referred to henceforth as $Casp8^{lox/lox}$ or control), apart from a minor yet significant reduction in % weight gain in $Casp8^{LysMcre}Ripk3^{-/-}$ mice (Figs 2A and S3A). In contrast, over 25 wk of HFD feeding, $Casp8^{LysMcre}Ripk3^{-/-}$ mice and RIPK3-deficient mice exhibited delayed weight gain, with $Casp8^{LysMcre}Ripk3^{-/-}$ mice gaining significantly less weight (% of body mass) compared with control mice (Figs 2B and S3B). This discrepancy in weight was not grossly attributable to altered food intake or fecal output (Fig S3C). After 25 wk on ND, there were no major differences in organ weights, including subcutaneous adipose tissue (SAT) and visceral adipose tissue (VAT), although both $Casp8^{lox/lox}Ripk3^{-/-}$ and $Casp8^{LysMcre}Ripk3^{-/-}$ animals tended to have smaller livers (Fig S3D, E, and G). Likewise, all HFD-fed mice displayed comparable enlargement of SAT and VAT (Figs 2C and S3F and G). Impaired weight gain in HFD-fed $Casp8^{LysMcre}Ripk3^{-/-}$ mice instead correlated with a significantly less liver enlargement, with RIPK3 deficiency also conferring protection (Figs 2C and S3D and G). No overall difference in weight gain or organ mass was observed in genetic control mice lacking caspase-8 only in myeloid cells ($Casp8^{LysMcre}$), compared with $Casp8^{lox/lox}$ control mice (Fig S3H and I) (Kang et al, 2004; Lawlor et al, 2015), highlighting that loss of caspase-8 may not be sufficient to trigger spontaneous RIPK3-mediated necroptotic inflammation in myeloid cells during obesity.

Examination of glycemic control revealed that ND- and HFD-fed $Casp8^{lox/lox}Ripk3^{-/-}$ and $Casp8^{LysMcre}Ripk3^{-/-}$ mice had comparable fasting blood glucose levels when compared to $Casp8^{lox/lox}$ controls (Figs 2D and E and S4A–C). ND-fed control and mutant animals also exhibited no significant difference in glucose clearance during an intraperitoneal glucose tolerance test (IP-GTT) after 8 or 16 wk of diet (Figs 2D and S4A). In contrast, whilst all HFD-fed mice developed progressive glucose resistance, $Casp8^{LysMcre}Ripk3^{-/-}$ mice demonstrated superior glucose clearance over time compared with control and RIPK3-deficient mice (Figs 2E and S4B and C), but were equally resistant to insulin after ~23 wk of HFD (Fig S4D). At end-point, HFD-fed $Casp8^{lox/lox}Ripk3^{-/-}$ and $Casp8^{LysMcre}Ripk3^{-/-}$ mice exhibited lower starved systemic insulin levels (Fig S4E and F), which may be indicative of better insulin sensitivity. Strikingly, examination of other systemic metabolic disease markers in ND- and HFD-fed mice (Fig S4E–J) revealed that HFD-fed $Casp8^{LysMcre}Ripk3^{-/-}$ mice, and to a slightly lesser extent $Casp8^{lox/lox}Ripk3^{-/-}$ mice, exhibited reduced signs of dyslipidemia, with lower serum triglyceride and cholesterol levels (Fig S4G and H). In contrast, serum ALT and AST levels (as markers of liver damage) were not significantly attenuated in the HFD-fed $Casp8^{LysMcre}Ripk3^{-/-}$ cohort (Fig S4I and J). Together, these data indicate that RIPK3 contributes to obesity-induced metabolic dysfunction, and this is likely, in part, through activation of caspase-8 signaling in myeloid cells.

## Loss of RIPK3 and caspase-8 activity in myeloid cells reduces local inflammation and steatosis

The accumulation of monocyte-derived CD11c+F4/80+ macrophages in adipose tissue that form crown-like structures around dying adipocytes is believed to drive the chronic low-level inflammation that leads to metabolic dysfunction and insulin resistance (Cinti et al, 2005). We therefore examined whether RIPK3 and caspase-8 contribute to pathological inflammatory changes in HFD-induced obesity. In line with comparable VAT weights between HFD-fed groups (Figs 2C and S3G), the mean adipocyte size from HFD-fed $Casp8^{lox/lox}Ripk3^{-/-}$ and $Casp8^{LysMcre}Ripk3^{-/-}$ mice was equivalent to control mice (Figs 2F and G and S5A and B). Yet, intriguingly, in the

**Figure 2. RIPK3 deficiency and myeloid-specific loss of caspase-8 reduce HFD-induced metabolic dysfunction, adipose tissue inflammation, and MAFLD development.**

**(A, B)** Body weights were measured weekly in $Casp8^{lox/lox}$ control (WT), $Casp8^{lox/lox}Ripk3^{-/-}$, and $Casp8^{LysMcre}Ripk3^{-/-}$ mice fed a (A) normal chow diet (ND) or (B) high-fat diet (HFD) for ~25 wk. Data shown are the mean ± SEM, n ≥ 11 ND-fed mice per group and n ≥ 13 HFD-fed mice per group pooled from three independent experiments. One-way ANOVA of the AUC, **P < 0.01. **(C)** End-stage organ weights in HFD-fed $Casp8^{lox/lox}$ control, $Casp8^{lox/lox}Ripk3^{-/-}$, and $Casp8^{LysMcre}Ripk3^{-/-}$ mice. Data shown are the mean ± SEM, n ≥ 11 mice per group pooled from three independent experiments. One-way ANOVA followed by Tukey's multiple comparison test, *P < 0.05. Gray boxes in (C) show the mean ± SEM from $Casp8^{lox/lox}$, $Casp8^{lox/lox}Ripk3^{-/-}$, and $Casp8^{LysMcre}Ripk3^{-/-}$ ND-fed mice (Fig S3D) for comparisons. **(D, E)** Glucose tolerance was measured in (D) ND- and (E) HFD-challenged $Casp8^{lox/lox}$ control, $Casp8^{lox/lox}Ripk3^{-/-}$, and $Casp8^{LysMcre}Ripk3^{-/-}$ mice by intraperitoneal glucose tolerance tests (IP-GTT; 1.5 g/kg) at 8 wk. Data shown are the mean ± SEM, n = 3–6 mice per group representative of 2–3 independent experiments. One-way ANOVA of the AUC, *P < 0.05. **(F)** Representative microscopy images of F4/80-immunostained VAT sections from ND- and HFD-fed $Casp8^{lox/lox}$ control, $Casp8^{lox/lox}Ripk3^{-/-}$, and $Casp8^{LysMcre}Ripk3^{-/-}$ mice at 25 wk. Arrows point to macrophage crown-like structures. The scale bar is 200 μm. **(G)** Mean adipocyte size in HFD-fed $Casp8^{lox/lox}$ control, $Casp8^{lox/lox}Ripk3^{-/-}$, and $Casp8^{LysMcre}Ripk3^{-/-}$ VAT was quantified using an automated algorithm on H&E-stained sections (Fig S5A). Data are the mean ± SEM, n ≥ 11 mice per group pooled from three independent experiments. One-way ANOVA followed by Tukey's multiple comparison test. **(H)** Numbers of neutrophils, inflammatory monocytes, and macrophages were quantified in the VAT of HFD-fed $Casp8^{lox/lox}$ control, $Casp8^{lox/lox}Ripk3^{-/-}$, and $Casp8^{LysMcre}Ripk3^{-/-}$ mice by flow cytometric analysis. Data shown are the mean ± SEM, n = 7–8 mice per group pooled from two independent experiments. One-way ANOVA followed by Tukey's multiple comparison test, *P < 0.05, **P < 0.01, ***P < 0.001. **(I)** VAT from HFD-fed $Casp8^{lox/lox}$ control, $Casp8^{lox/lox}Ripk3^{-/-}$, and $Casp8^{LysMcre}Ripk3^{-/-}$ mice was cultured ex vivo with and without LPS (50 ng/ml) overnight, and IL-1β and TNF were measured in the supernatants by ELISA. Data shown are the mean ± SEM, n ≥ 12 mice per group pooled from three independent experiments. One-way ANOVA followed by Dunnett's multiple comparison test, *P < 0.05, **P < 0.01. **(J)** Representative microscopy images of H&E-stained liver sections. The scale bar is 100 μm. **(K)** Histopathological evaluation of MAFLD in $Casp8^{lox/lox}$ control, $Casp8^{lox/lox}Ripk3^{-/-}$, and $Casp8^{LysMcre}Ripk3^{-/-}$ ND-fed and HFD-fed mice after 25 wk of challenge. Data shown are the mean ± SEM, n ≥ 9 mice per group pooled from three independent experiments. One-way ANOVA followed by Tukey's multiple comparison test, *P < 0.05, **P < 0.01.
Source data are available for this figure.

VAT of HFD-fed *Casp8*[LysMcre]*Ripk3*[−/−] mice, and to a lesser degree *Casp8*[lox/lox]*Ripk3*[−/−] mice, less inflammation was observed, as evidenced by fewer F4/80[+] crown-like structures with signs of apoptotic cleaved caspase-3 activity (Figs 2F and S5A), less CD11c[+]MHCII[hi] macrophage, monocyte, and neutrophil infiltrate (Figs 2H and S5C), and reduced secretion of TNF and NLRP3 inflammasome–activated IL-1β (to LPS treatment) (Fig 2I). Histo-pathological analysis of the livers also revealed that RIPK3 defi-ciency and myeloid-specific caspase-8 loss diminished hepatic steatosis, ballooning, and fibrosis in HFD-fed (and in ND-fed) mice (Figs 2J and K and S5D), whereas flow cytometric analysis suggested a trend toward reduced recruitment of inflammatory neutrophils and monocyte/macrophages in *Casp8*[lox/lox]*Ripk3*[−/−] and *Casp8*[LysMcre]*Ripk3*[−/−] HFD-fed mice, compared with controls (Fig S5E–G). Correlating with this was a reduction in the expression of inflammatory genes, including NLRP3 inflammasome machinery (Fig S5H). Collectively, these results together with our in vitro findings (Fig 1) suggest that RIPK3 and myeloid cell caspase-8 activity regulates adipose tissue inflammation and the progres-sion to MAFLD/MASH.

### MLKL drives obesity-induced metabolic dysfunction and insulin resistance

We next examined the effect of the RIPK3 substrate MLKL on the development of obesity and MAFLD. ND-fed MLKL-deficient mice displayed steady weight gain, although weights peaked lower than WT control mice (Figs 3A and S6A) and were associated with a reduction in SAT and VAT weights (Figs 3B and S6C and E). Im-pressively, HFD-fed *Mlkl*[−/−] mice exhibited markedly diminished weight gain over time, compared with WT controls (Figs 3C and S6B), which correlated with less SAT, VAT, and liver tissue expansion (Figs 3D and S6D and E), and not with reduced food intake or increased fecal output (Fig S6F). Upon testing glucose and insulin resistance over time, *Mlkl*[−/−] mice demonstrated less metabolic dysfunction on a HFD, and to a lesser extent on a ND (Fig 3E-I and S6G). Analysis of metabolic and damage markers in ND-fed mice revealed that fasting serum glucose, insulin, ALT, AST, triglyceride, cholesterol, and non-esterified fatty acid (NEFA) levels were largely normal in *Mlkl*[−/−] mice (Fig 3J–O, top panel, and Fig S6H). However, upon HFD feeding, *Mlkl*[−/−] mice exhibited lower serum insulin, ALT, and cholesterol levels, compared with HFD-fed WT controls (Fig 3J–O, bottom panel, and Fig S6H), supporting the idea that MLKL drives obesity-induced metabolic dysfunction.

### MLKL regulates adiposity and the development of MAFLD

We next examined VAT and liver pathology to discern how MLKL drives obesity and liver disease. Correlating with lower VAT weights (Fig 3B and D), ND- and HFD-fed *Mlkl*[−/−] adipocytes were smaller on average (Fig 4A and B), suggesting that MLKL regulates adiposity. Surprisingly, further analysis of inflammation in the VAT from HFD-fed mice revealed that although fewer F4/80[+] crown-like macro-phage structures were present in the VAT of MLKL-deficient mice, compared with WT controls (Figs 4A and S6I), there was only a trend toward less inflammatory cell infiltrate (Fig 4C). Correspondingly, LPS-induced IL-1β and TNF secretion in the VAT was only modestly

impaired (Fig 4D). Analysis of liver pathology also revealed that MLKL-deficient mice were markedly protected from steatosis and ballooning from HFD feeding, with a similar trend in aging ND-fed mice (Figs 4E and F and S6J). *Mlkl*[−/−] livers also exhibited a lower proportion of monocyte/macrophages within the liver tissue (Figs 4G and S5E), and this reduction was associated with reduced *Tnf* expression and a trend toward lower levels of the monocyte-attracting chemokine *Ccl2*, but not inflammasome-associated genes *Nlrp3* and *Il1b* (Fig 4H). Therefore, MLKL drives obesity and MAFLD in aged and HFD-fed mice but does not contribute sub-stantially to inflammation.

To evaluate whether hematopoietic MLKL expression contributes to obesity, we examined C57BL/6 mice reconstituted with WT control or *Mlkl*[−/−] bone marrow in our dietary model. Both weekly weighing and measurement of fat mass using an EchoMRI in-strument highlighted that WT and *Mlkl*[−/−] bone marrow chimeras responded equivalently to ND or HFD feeding (Fig S7A and B). Furthermore, in contrast to global MLKL-deficient mice, no differ-ences in end-stage SAT, VAT, or liver weights were observed in chimeric animals (Fig S7C) nor were there any signs of improved glucose metabolism upon an oral GTT (Fig S7D). These results, combined with the distinct phenotype of adipocyte hypertrophy and fatty liver damage observed in aged or HFD-fed MLKL-deficient mice, suggest that MLKL alters tissue homeostasis to cause obesity-induced metabolic disease.

### MLKL signaling induces a lipid metabolic gene signature in the liver of aging and obese mice

In view of the protection from MAFLD and metabolic dysfunction observed in MLKL-deficient mice, we next examined global hepatic responses in aged ND-fed or HFD-fed mice. 3′ mRNA sequencing of WT and *Mlkl*[−/−] livers revealed that there were significant tran-scriptomic changes in differentially expressed genes between genotypes and diets (Fig 5A and B). In WT HFD-fed mice, more than 1,700 genes (*P* < 0.05) were up- and down-regulated in the liver compared with WT ND-fed animals, whereas *Mlkl*[−/−] HFD mice exhibited 1,111 up-regulated and 948 down-regulated genes, compared with knockout ND mice (*P* < 0.05). As expected, after HFD feeding, gene ontology (GO) analyses revealed up-regulation of metabolic and inflammatory processes in WT livers, such as cho-lesterol biosynthesis, acyl-CoA metabolic process, peroxisome proliferator–activated receptor (PPAR) signaling pathway, and in-flammatory response (Fig S8A). Likewise, livers from HFD-fed versus ND-fed *Mlkl*[−/−] mice showed up-regulation of lipid metabolism–associated terms (e.g., fatty acid metabolic process, long-chain fatty acid metabolic process) (Fig S8B). Importantly, direct comparisons between WT and *Mlkl*[−/−] livers revealed that MLKL deficiency in-duces significant changes in differentially expressed genes in HFD mice, and modest changes in ND (Fig 5B and C, S8C–E, and S9).

Gene set enrichment analysis (GSEA) and GO analysis revealed that in the absence of MLKL, the expression of genes associated with cholesterol biosynthesis and homeostasis was increased in the aging liver (Figs 5C and S8C–E). However, qRT–PCR analysis of key cholesterol metabolism regulatory genes (*Srebp2*, *Hmgcr*, *Hmgcs2*, and *Ldlr*) showed they were not significantly altered in ND-

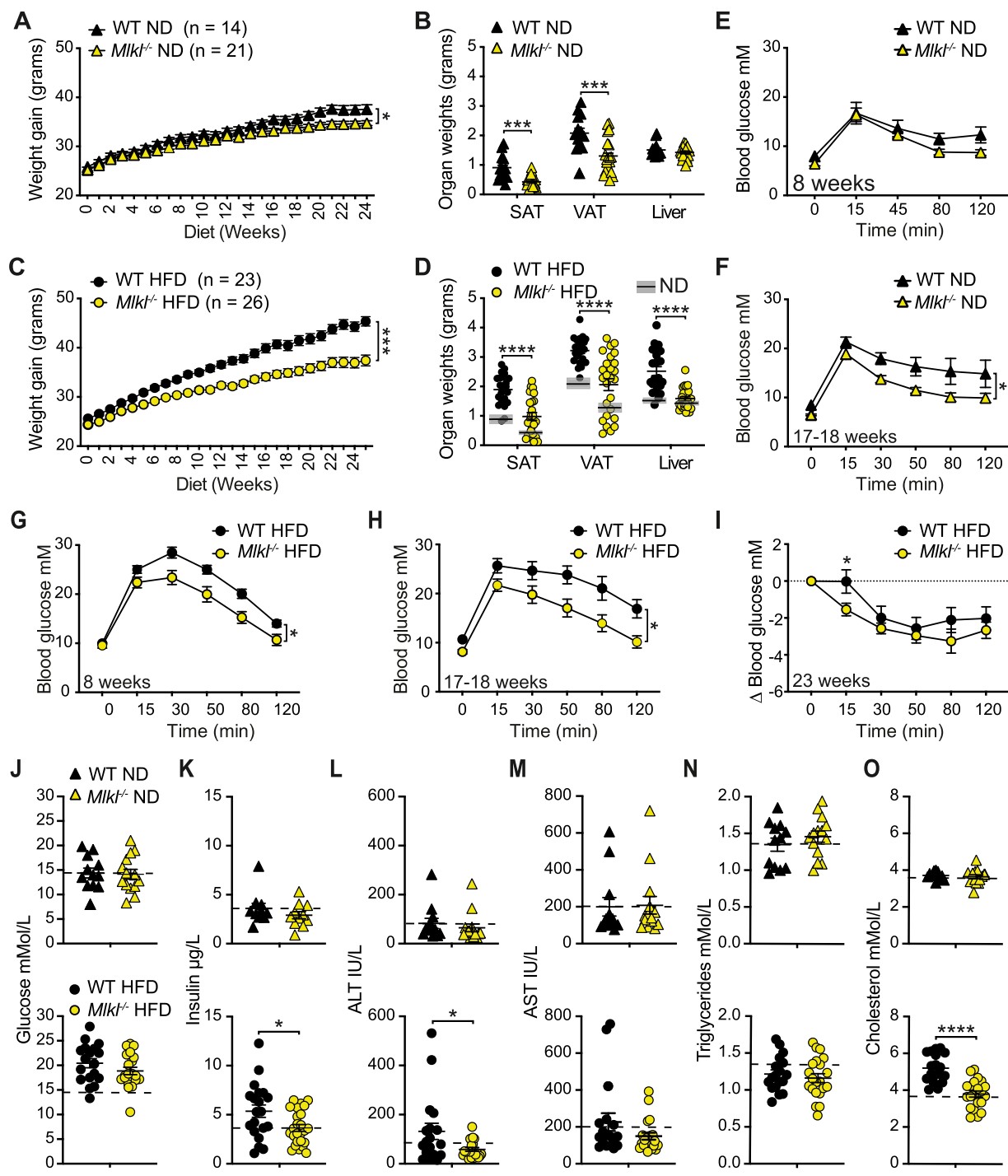

**Figure 3. *Mlkl*$^{-/-}$ mice display reduced obesity and metabolic dysfunction with aging and with HFD feeding.**
**(A, C)** WT and *Mlkl*$^{-/-}$ mice were fed an (A) ND and (C) HFD for ~24–25 wk and body weights measured on a weekly basis. Data shown are the mean ± SEM, n ≥ 14 ND-fed mice/group and n ≥ 23 HFD-fed mice/group pooled from three independent experiments. Unpaired, two-tailed *t* test of the AUC, \*$P$ < 0.05, \*\*\*$P$ < 0.001. **(B, D)** End-stage organ weights from (B) ND- and (D) HFD-fed WT and *Mlkl*$^{-/-}$ mice. Data shown are the mean ± SEM, n ≥ 14 mice/group pooled from three independent experiments. Unpaired, two-tailed *t* test, \*\*\*$P$ < 0.001, \*\*\*\*$P$ < 0.0001. Gray boxes in (D) represent the mean ± SEM from (B) for comparison. **(E, F, G, H)** Glucose tolerance was assessed via an IP-GTT (1.5 g/kg) at 8 and 17–18 wk in ND- and HFD-fed WT and *Mlkl*$^{-/-}$ mice. Data shown are the mean ± SEM, (E, F) n = 5–6 ND-fed mice per group and (G, H) n = 8–9 HFD-fed mice per group and are representative of one of 2–3 experiments. Unpaired, two-tailed *t* test of the AUC, \*$P$ < 0.05. **(I)** Insulin resistance was assessed in HFD-fed WT and *Mlkl*$^{-/-}$ mice (~23 wk) during an IP-ITT (0.75 U/kg). Data shown are the mean ± SEM, n = 8–9 mice/group from one of two experiments. Unpaired, two-tailed *t* test of the AUC or independent time points, \*$P$ < 0.05. **(J, K, L, M, N, O)** Fasting serum, (J) glucose, (K) insulin, (L) ALT, (M) AST, (N) triglyceride, and (O) cholesterol after 24–25 wk of ND or HFD feeding in WT and *Mlkl*$^{-/-}$ mice. Data shown are the mean ± SEM, n ≥ 10 ND-fed mice per group (top panel) and n ≥ 18 HFD-fed mice per group (bottom panel). Unpaired, two-tailed *t* test, \*$P$ < 0.05, \*\*\*\*$P$ < 0.0001.
Source data are available for this figure.

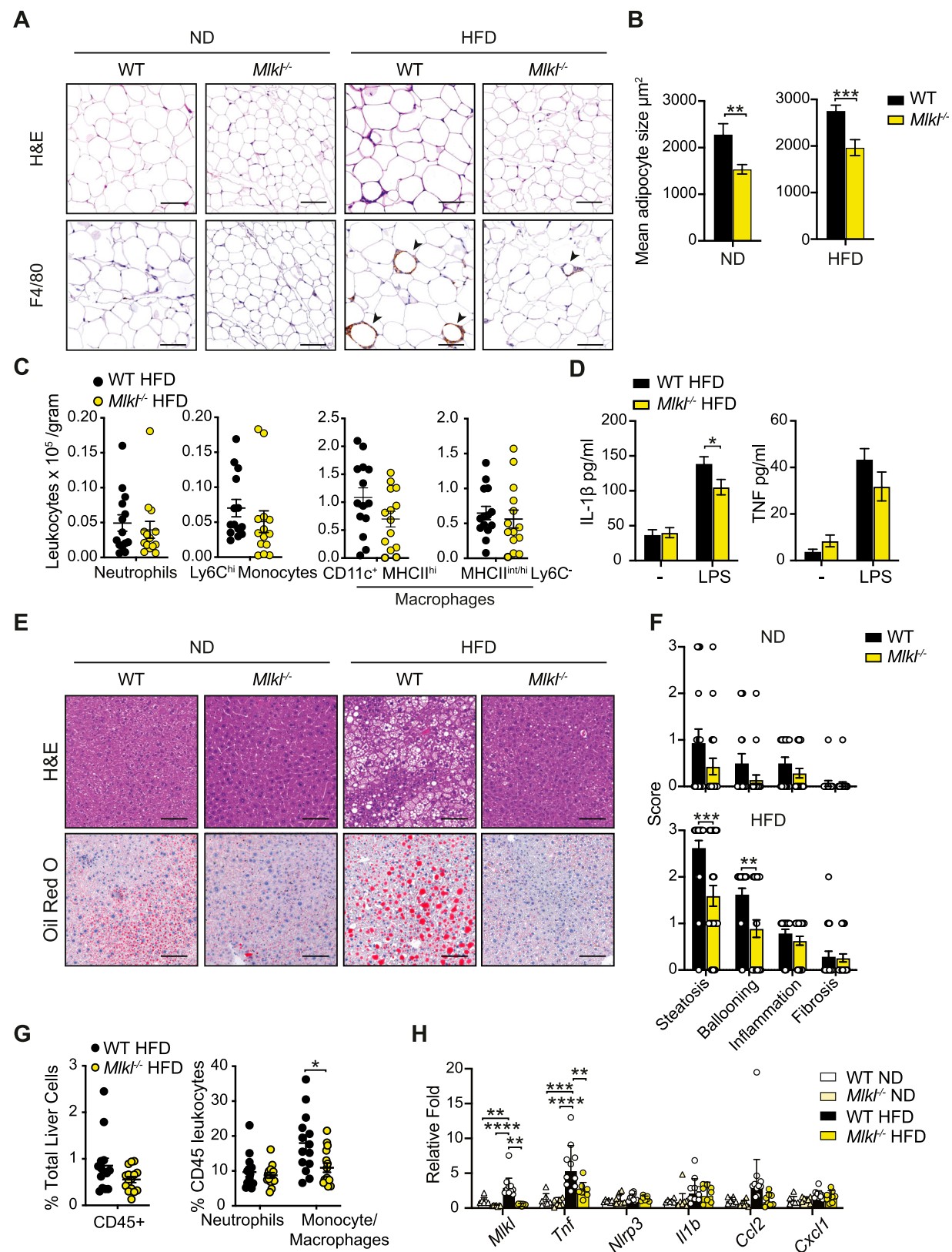

**Figure 4. MLKL deficiency reduces adiposity and fatty liver disease in response to HFD challenge.**
**(A)** Representative microscopy images of H&E-stained and F4/80-immunostained VAT sections from WT and *Mlkl*[−/−] mice fed a ND or HFD for ~25 wk. Arrows point to crown-like structures. The scale bar is 100 *μm*. **(B)** Automated quantification of the mean adipocyte size in VAT on H&E-stained sections (A). Data shown are the mean ±

or HFD-fed $Mlkl^{-/-}$ mice (Fig S10A). Conversely, GO analyses revealed that ND-fed $Mlkl^{-/-}$ livers exhibited down-regulated gene sets for various lipid-associated and metabolic processes, including biosynthesis of unsaturated fatty acids and positive regulation of the lipid metabolic process (Fig 5C). GSEA further supported this trend, showing down-regulation of genes associated with oxidative phosphorylation, fatty acid metabolism, peroxisomes, and adipogenesis (Fig S8D). Importantly, HFD-fed $Mlkl^{-/-}$ mice did not up-regulate these gene signatures, apart from oxidative phosphorylation that was up-regulated compared with HFD-fed WT livers and characterized by increased electron transport chain gene expression (Figs S8D and E and S9A). GO analysis further complemented this pattern with down-regulation of gene sets involved in several lipid and membrane regulatory/signaling processes in HFD-fed $Mlkl^{-/-}$ livers, such as regulation of cytokine production, positive regulation of lipid localization, positive regulation of lipid biosynthetic process, and PPAR signaling pathway (Figs 6A and S9B–D). Notably, many genes down-regulated in HFD MLKL–deficient livers that were involved in regulation of cytokine production were extracellular or intracellular sensors/ receptors that coordinate cell signaling responses (Fig S9B).

Comparisons of down-regulated lipid/membrane-related GO terms in HFD-fed $Mlkl^{-/-}$ livers revealed an overlap in genes regulated by nuclear PPAR signaling (Figs 5D and 6A and S9B–D). Correspondingly, qRT–PCR analysis validated that several PPAR-related genes that were differential in GO analyses (Figs 6A and S9B–D) and/or are associated with key lipogenic processes were reduced in $Mlkl^{-/-}$ livers with HFD feeding (and trended down in ND-fed animals), including the transcription factors *Ppara/g* themselves, as well as genes associated with fatty acid uptake (*Cd36*, *Fabp2/4*, *Vldlr*), fatty acid synthesis (*Acaca*, *Fasn*, *Srebp1*), elongation (*Elovl6*) and desaturation (*Scd1*), triglyceride synthesis (*Mogat1*, *Dgat1*), peroxisome function (*Acot3*, *Abcd2*, *Acaa1b*), lipid droplet storage (*Plin4*), and death/lipolysis (*Cidea*) (Figs 6B–G and S10B). As CD36, a key membrane fatty acid translocase, is important for fatty acid uptake and can trigger a PPAR-regulated positive feedback loop and several other lipid homeostatic processes (Rada et al, 2020; Hajri et al, 2021; Zeng et al, 2022), we also assessed CD36 levels by immunoblot and uncovered that unlike WT livers, HFD-fed $Mlkl^{-/-}$ livers did not up-regulate CD36 protein levels (Fig 6H). Likewise, we also observed lower levels of the intracellular fatty acid transporter FABP4 in aging ND- and HFD-fed $Mlkl^{-/-}$ livers, when compared to WT mice (Fig S10C). These findings suggest that MLKL regulates molecules involved in lipid uptake/transport, synthesis, and signaling.

Based on the differing obesity and inflammatory phenotypes observed in RIPK3- versus MLKL-deficient mice, we questioned whether MLKL requires RIPK3 activity to alter lipid metabolism in the liver of HFD-fed mice. Interestingly, although immunoblot analysis revealed that RIPK1 and MLKL are expressed in the liver, and up-regulated in WT mice with a HFD, neither RIPK3 nor phosphorylated MLKL (active) was detectable (Fig S11A). Immunostaining confirmed that RIPK3 was not detectable in liver hepatocytes and that its expression was largely restricted to macrophages and crown-like structures surrounding lipid droplets and adipocytes in both the liver and VAT, respectively (Fig S11B). As RIPK3 levels may be below the threshold of detection, we next chose to analyze the effects of saturated fatty acid palmitate on RIPK3 expression and function in vitro using WT and $Mlkl^{-/-}$ Hepa1-6 hepatic cell lines that display epigenetic silencing of RIPK3 (Fig S12A) (Preston et al, 2022), rendering them responsive to TNF-induced apoptosis (via treatment with TNF and Smac mimetic, TS), but resistant to TNF-induced necroptosis (TS and Q-VD-OPh treatment, TSQ) (Fig S12B). Importantly, reconstitution of these cells with doxycycline (DOX)-inducible RIPK3 allowed restoration of RIPK3 signaling and provided a model system where WT Hepa1-6 cells but not $Mlkl^{-/-}$ cells were sensitive to TSQ killing (Fig S12A and B). Examination of lipid responses using this system revealed that not only did high doses of palmitate fail to induce RIPK3 expression in either WT or $Mlkl^{-/-}$ Hepa1-6 cells (Fig S12C), but palmitate-induced cell death at 16 h did not require RIPK3 and MLKL activity (Fig S12C and D). Strikingly, however, lipid accumulation, as measured by BODIPY staining, was reduced in $Mlkl^{-/-}$ Hepa1-6 cells to low (less lipotoxic) doses of palmitate, although again this was independent of RIPK3 expression (Fig S12E), suggesting that MLKL may noncanonically impact lipid uptake and intracellular signaling in hepatic cells. Strengthening this concept, $Mlkl^{-/-}$ Hepa1-6 cells appeared to have reduced basal- or palmitate-induced expression of several lipid metabolism–related genes, compared with WT cells (Fig S12F–K). Overall, these results suggest that MLKL may regulate lipid uptake and transcriptional responses independent of RIPK3 in the liver.

### MLKL regulates the synthesis of monounsaturated and polyunsaturated diglycerides and triglycerides in the liver

The N-terminal four-helical bundle domain of MLKL dominantly binds negatively charged phosphatidylinositol phosphates, particularly PI(4,5)P$_2$, in the plasma membrane to facilitate necroptotic

SEM, n ≥ 13 ND-fed mice per group and n ≥ 23 HFD-fed mice per group pooled from three independent experiments. Unpaired, two-tailed *t* test, **$P < 0.01$, ***$P < 0.001$. **(C)** VAT from HFD-fed WT and $Mlkl^{-/-}$ mice was harvested at ~25 wk, and the numbers of neutrophils, inflammatory monocytes, and macrophages were quantified by flow cytometric analysis. Data shown are the mean ± SEM, n ≥ 14 mice per group pooled from two independent experiments. Unpaired, two-tailed *t* test. **(D)** VAT from HFD-fed WT and $Mlkl^{-/-}$ mice was harvested at ~25 wk and cultured ex vivo with and without LPS (50 ng/ml) overnight, and IL-1β and TNF were measured in the supernatants by ELISA. Data shown are the mean ± SEM, n ≥ 19 mice per group from pooled from three independent experiments. Unpaired, two-tailed *t* test, *$P < 0.05$. **(E)** Representative microscopy images of H&E–stained and Oil Red O–stained (to detect lipid droplets) liver sections. The scale bar is 100 µm. **(F)** Histopathological evaluation of disease in WT and $Mlkl^{-/-}$ mice after 25 wk of ND or HFD feeding. Data shown are the mean ± SEM, n ≥ 14 ND-fed mice and n ≥ 23 HFD-fed mice pooled from three independent experiments. Unpaired, two-tailed *t* test, **$P < 0.01$, ***$P < 0.001$. **(G)** Flow cytometric analysis of the proportion of CD45+ leukocytes in the livers of WT and $Mlkl^{-/-}$ HFD mice that are neutrophils and monocyte/macrophages. Data shown are the mean ± SEM, n ≥ 14 mice per group pooled from two independent experiments. Unpaired, two-tailed *t* test, *$P < 0.05$. **(H)** qRT–PCR measurement of relative *Mlkl*, *Tnf*, *Nlrp3*, *Il1b Ccl2*, and *Cxcl1* mRNA expression in ND- and HFD-fed WT and $Mlkl^{-/-}$ liver tissues after 23–25 wk of diet (fold change over WT ND). Data shown are the mean ± SEM, n ≥ 6 mice per group pooled from three experiments. Unpaired, two-tailed *t* test, **$P < 0.01$, ***$P < 0.001$, ****$P < 0.0001$.
Source data are available for this figure.

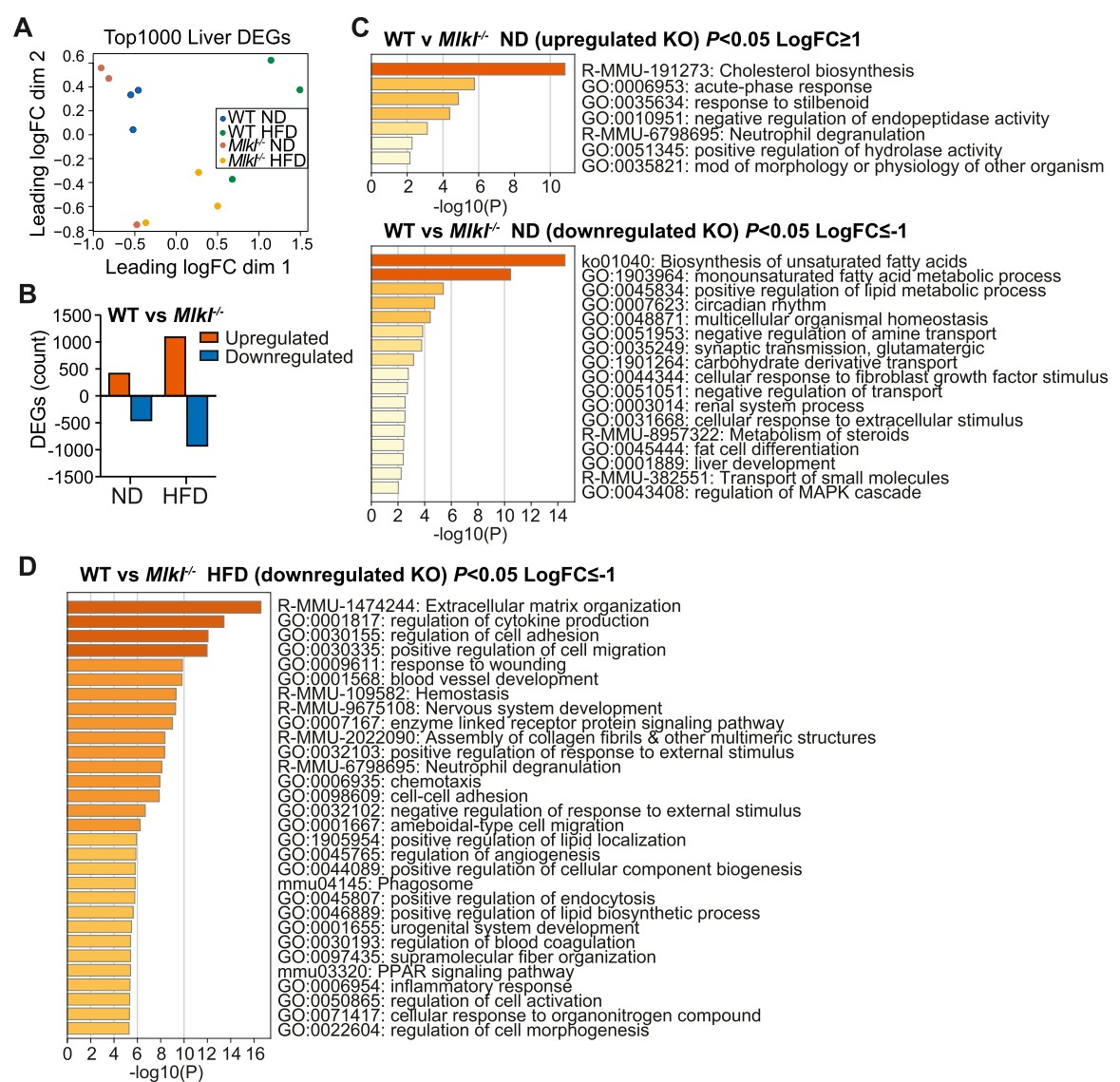

**Figure 5. Deficiency in MLKL leads to altered gene signatures in the livers of aging and HFD-fed mice.**
**(A, B, C, D)** Liver RNA extracted from WT and $Mlkl^{-/-}$ mice on a ND or HFD (n = 3 mice per group) was subjected to 3′ mRNA sequencing. (A) Multidimensional scaling plot and (B) the number of differentially expressed genes up-regulated and down-regulated in WT v $Mlkl^{-/-}$ ND and HFD livers. $P ≤ 0.05$ and cutoff values logFC ≥ 1 or logFC ≤ −1. (C) Gene ontology (GO) pathways of significant DEGs up-regulated and down-regulated in $Mlkl^{-/-}$ ND livers compared with WT ND livers. $P ≤ 0.05$ and cutoff values logFC ≥ 1 or logFC ≤ −1. (D) Top 30 GO pathways of significant DEGs down-regulated in $Mlkl^{-/-}$ HFD livers compared with WT HFD livers. $P ≤ 0.05$ and cutoff values logFC ≤ −1. Source data are available for this figure.

death (Wang et al, 2014; Quarato et al, 2016; Sethi et al, 2022). Beyond this, necroptosis activation and signaling may be tightly regulated by lipid species (Zhang et al, 2020), as saturated very long-chain fatty acids and acylation of phospho-MLKL/MLKL are required to promote endocytic trafficking of MLKL to the membrane (Parisi et al, 2019; Pradhan et al, 2021). Based on the perturbed lipid metabolic/membrane receptor gene expression signatures in the livers of MLKL-deficient mice, which appear to be independent of RIPK3, we sought to understand how MLKL signaling impacts the abundance of individual lipid species in the serum, VAT, and liver of aging and HFD-fed mice using targeted lipidomic analysis. Lipid metabolic clustering was primarily associated with diet, but some divergence of serum and liver profiles between genotypes was observed on

HFD, with a modest shift in hepatic lipids also seen with ND feeding (Fig S13A). In contrast, based on the PCA plot, only a minor shift in the VAT lipidome was observed in HFD-fed WT and MLKL-deficient mice (Fig S13A).

Analysis of the relative abundance of individual lipid classes in the serum, liver, and VAT of aging ND-fed WT and $Mlkl^{-/-}$ mice revealed no major changes (Figs 7A and B and S13A). No gross perturbation in the lipid classes was also observed in the VAT of HFD-fed WT and $Mlkl^{-/-}$ mice, although specific analysis of obesity-associated ceramide (Cer) species revealed a downward trend in the relative abundance of most species in HFD-fed $Mlkl^{-/-}$ VAT, including toxic long-chain fatty acid species (Fig S13B) (Chaurasia et al, 2020). Analysis of the relative abundance

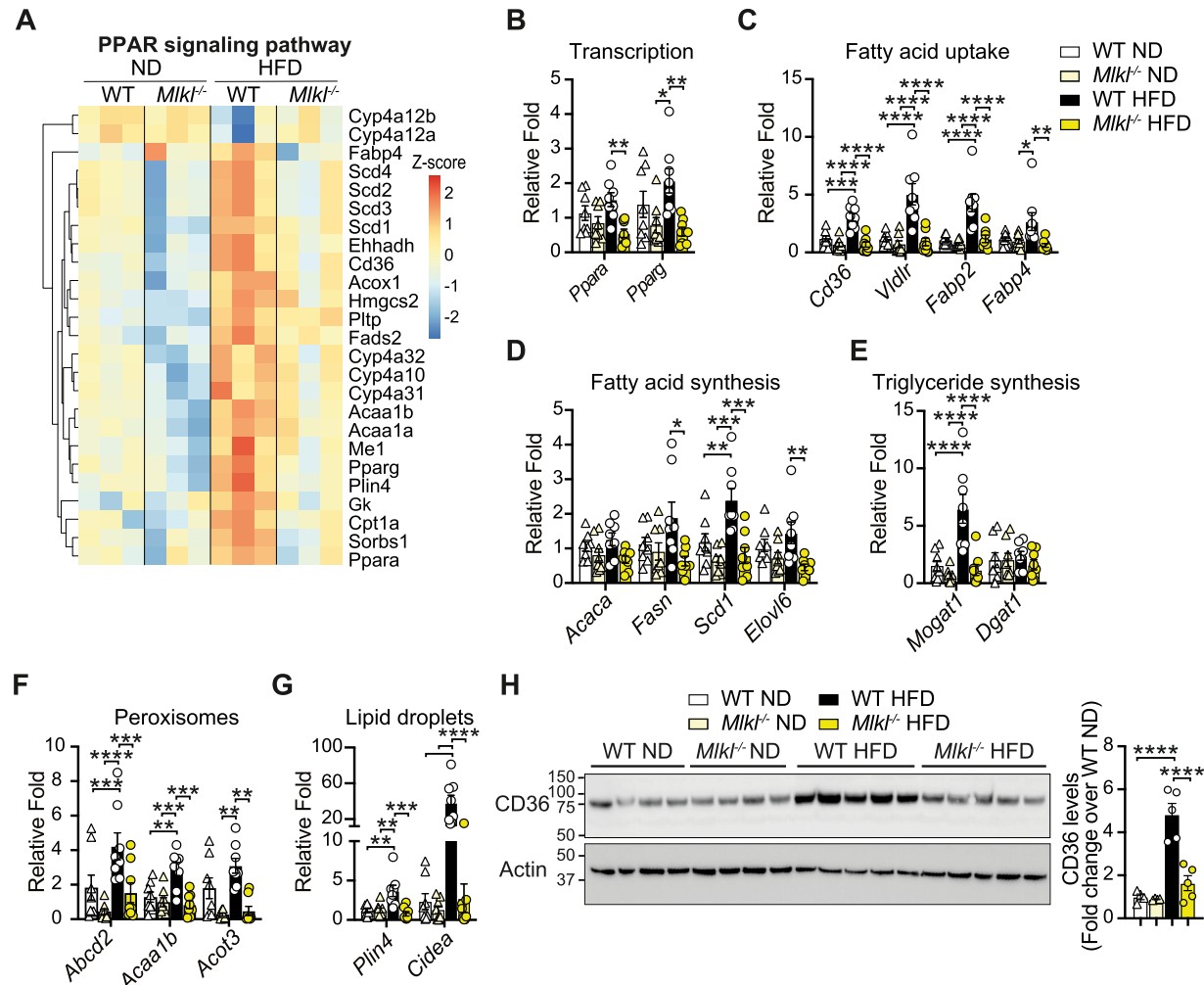

**Figure 6. PPAR signaling–associated lipid metabolism genes are down-regulated in the livers of HFD-fed *Mlkl*$^{-/-}$ mice.**
**(A)** Heatmap of significant DEGs of GO-term PPAR signaling pathway (down-regulated in *Mlkl*$^{-/-}$ HFD livers compared with WT HFD livers). $P \leq 0.05$ and cutoff values logFC $\geq 1$ or logFC $\leq -1$. **(B, C, D, E, F, G)** qRT–PCR analysis of liver mRNA from WT and *Mlkl*$^{-/-}$ mice fed a ND or HFD. Data shown are the mean ± SEM, n = 6–8 mice per group pooled from three independent experiments. Unpaired, two-tailed t test, *$P < 0.05$, **$P < 0.01$, ***$P < 0.001$, ****$P < 0.0001$. **(H)** Liver lysates from ND- and HFD-fed WT and *Mlkl*$^{-/-}$ mice were analyzed by immunoblot for the indicated antibodies. n = 4–5 mice per group; each lane represents an individual mouse. CD36 levels were analyzed by densitometry and normalized to actin and expressed as a fold change over WT ND liver lysates. Results are presented as the mean ± SEM. One-way ANOVA followed by Tukey's multiple comparison test, ****$P < 0.0001$.
Source data are available for this figure.

of lipid classes in the serum of HFD-fed WT and *Mlkl*$^{-/-}$ mice also revealed no overall changes in total diglycerides (DG) and triglycerides (TG) (Fig 7A). However, levels of select monounsaturated fatty acid (MUFA) and polyunsaturated fatty acid (PUFA) TG species were significantly elevated, and a trend toward increased saturated TG was also evident (Figs 7C and S13C), perhaps indicative of active secretion and/or reduced uptake by metabolic tissues. Intriguingly, membrane lipids (e.g., sphingomyelin [SM], phosphatidylcholine [PC], and phosphatidylinositol [PI]), particularly those comprised of long to very long acyl chains with at least one double bond, were reduced in HFD-fed *Mlkl*$^{-/-}$ serum (Fig 7A and C).

Liver lipid profiling revealed that levels of DG were diminished in *Mlkl*$^{-/-}$ livers upon HFD feeding (Figs 7B and D and S13D). Correspondingly, pools of TG species and phospholipids PC, PI, PS, and

phosphatidylglycerol (PG), which are dependent on DG for their synthesis, were significantly attenuated in HFD-fed *Mlkl*$^{-/-}$ livers, as were SM species (Fig 7B and D). Closer examination of the DG and TG composition revealed lower abundance of saturated TG comprising three palmitic acids (C16:0) in *Mlkl*$^{-/-}$ livers, and a diminished abundance of MUFA or PUFA DG, TG, and phospholipid species containing at least one long-chain (oleic acid C18:1, arachidonic acid C20:4) or very-long-chain (nervonic acid C24:1, docosahexaenoic acid C22:6) fatty acid (Figs 7D and S13D). This pattern suggests that MLKL-deficient mice may be defective in the uptake and/or synthesis/remodeling of unsaturated long and very long acyl chain fatty acids. Moreover, these results correlate strongly with our RNA-seq gene signatures and in vitro assays that point to MLKL being a regulator of lipid homeostasis to drive obesity and fatty liver disease.

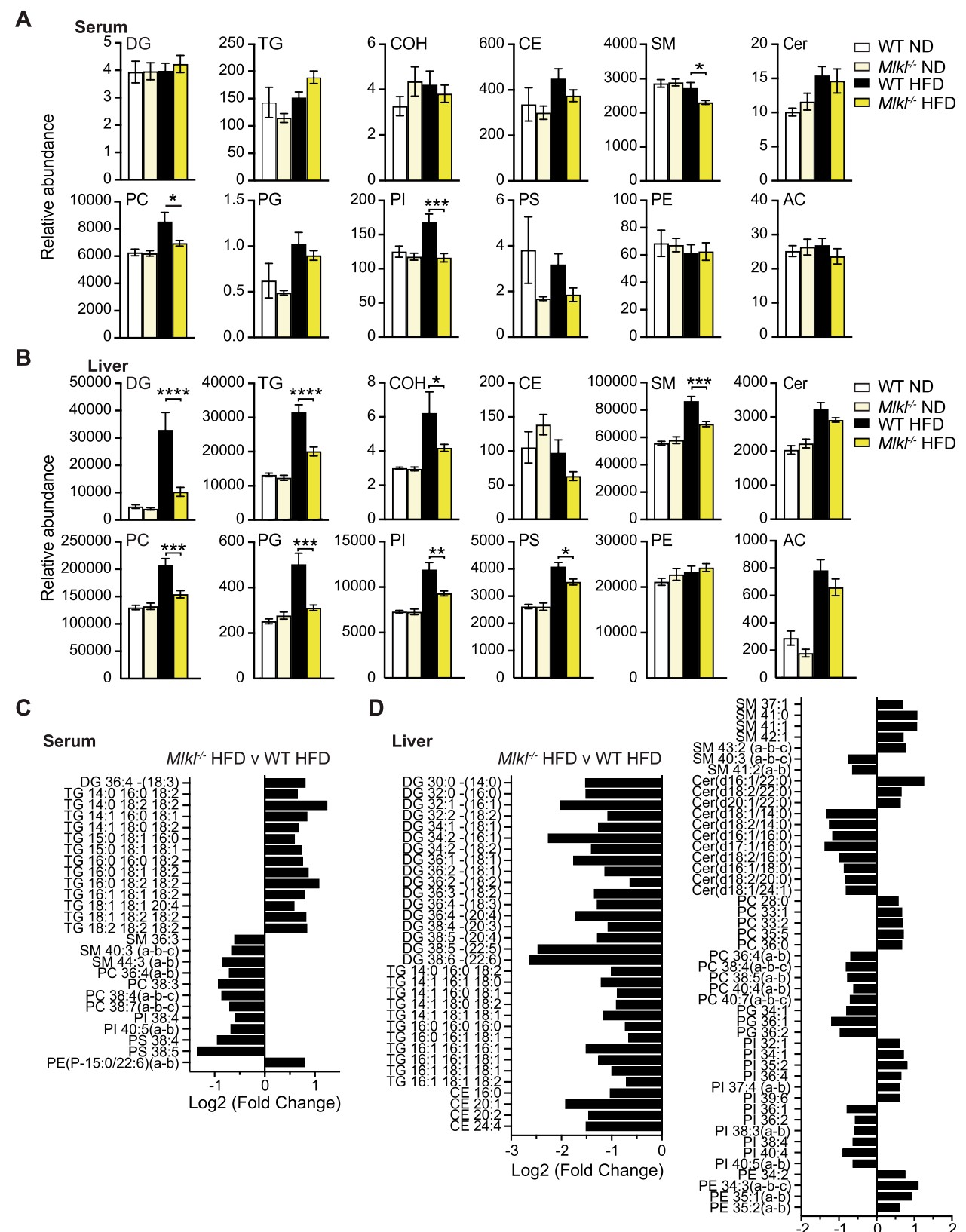

**Figure 7. MLKL deficiency alters the serum and liver lipid profile upon HFD feeding.**
**(A, B, C, D)** WT and $Mlkl^{-/-}$ were fed a normal chow diet (ND) or high-fat diet (HFD) for 25 wk. The total lipid was extracted from the serum, liver, and VAT, and lipid species were analyzed by LC-MS. **(A, B)** Relative abundance of total lipid classes in the (A) serum and (B) liver of mice. Data are normalized for the median lipid content per sample.

# Discussion

In this study, we genetically examined the contribution of extrinsic cell death signaling to HFD-induced obesity and associated metabolic dysfunction. Our findings show that extrinsic RIPK3/caspase-8 activity in macrophages largely induces chronic metabolic inflammation by regulating the transcriptional responses needed for efficient NLRP3 inflammasome activity. In comparison, MLKL appears to drive obesity with aging to promote MAFLD development in a manner independent of canonical RIPK3 signaling. Remarkably, the protection from MAFLD we observed in MLKL-deficient mice appeared to be conferred by reduced signals for lipid uptake, de novo lipogenesis, and triglyceride synthesis, and culminated in a deficit in long-chain and very-long-chain unsaturated lipid species.

The NLRP3 inflammasome has emerged as a major therapeutic target to limit pro-inflammatory IL-1$\beta$ activity in obesity-associated metabolic disorders (Coll et al, 2022). A large body of evidence places RIPK1/3-driven caspase-8 or MLKL signaling upstream of NLRP3 inflammasome activity in a range of disease settings (Gurung et al, 2014; Orning et al, 2018; Sarhan et al, 2018; Polykratis et al, 2019; Huang et al, 2021; Kim et al, 2022), yet their contribution to inflammasome activity in metabolic disease is less clear (Tao et al, 2021). Our findings reveal that although RIPK3 loss and myeloid-specific caspase-8 deletion mostly phenocopy the protection from adipose tissue inflammation, metabolic dysfunction, and steatohepatitis observed in NLRP3 inflammasome–deficient mice (Stienstra et al, 2010, 2011; Vandanmagsar et al, 2011), it failed to prevent adiposity and end-stage insulin resistance. Accordingly, we uncovered that neither MLKL nor caspase-8 signaling is obligatory for NLRP3 inflammasome activation in LPS-treated macrophages exposed to palmitate, but instead caspase-8 regulates inflammatory gene transcription, direct IL-1$\beta$ proteolysis, and to a lesser extent cell death. Regarding the latter, it is probable that saturated fatty acid crystallization (Karasawa et al, 2018) leads to caspase-independent lysosomal membrane rupture and cellular demise, as observed for other crystalline NLRP3-activating stimuli (Rashidi et al, 2019). Overall, our results suggest that targeting the apoptotic RIPK3/caspase-8 signaling axis may potentially limit myeloid cell–driven tissue inflammation and subsequently dampen MAFLD progression. However, the utility of this approach may be limited given the poor clinical performance of the pan-caspase inhibitor emricasan in MAFLD/MASH (Lekakis & Cholongitas, 2022).

Although the necroptotic kinases, RIPK1 and RIPK3, have been studied in liver damage models (Li et al, 2021), it remains controversial as to whether antagonizing these kinases and necroptotic MLKL signaling is a valid therapeutic option in obesity-driven MAFLD/MASH, as disease outcomes differ based on the dietary intervention and genetic models used (Gautheron et al, 2014, 2016; Afonso et al, 2015, 2021; Roychowdhury et al, 2016; Saeed et al, 2018;

Karunakaran et al, 2020; Tao et al, 2021; Pistorio et al, 2022). For example, RIPK3 deficiency partially protects mice from MASH development on a MCD diet or choline-deficient diet, but not a HFD (Gautheron et al, 2014; Afonso et al, 2015, 2021; Roychowdhury et al, 2016), whereas MLKL deficiency is dominantly reported to protect mice from diet-induced MAFLD (Saeed et al, 2019; Xu et al, 2019; Wu et al, 2020; Preston et al, 2022). In our study, we found that either RIPK3 or MLKL loss impedes MAFLD development with HFD feeding, but MLKL deletion also reduced adipose tissue hypertrophy and afforded greater protection from metabolic dysfunction, insulin resistance, and MAFLD. In comparison, MLKL deficiency more modestly attenuated inflammatory changes in VAT and liver tissue (i.e., TNF induction, inflammatory macrophage infiltration) compared with RIPK3-deficient mice and other studies in $Mlkl^{-/-}$ mice (Saeed et al, 2019; Wu et al, 2020). The reason for discrepancies between RIPK3 and MLKL knockout studies is unclear but is likely to be influenced by the metabolic shift caused by the dietary model adopted and duration of challenge, genetic background and use of littermates, and environmental variations (e.g., microbiome, housing).

Confounding the canonical model that RIPK3/MLKL-mediated necroptosis causes MAFLD, it has recently been shown that necroptosis in hepatocytes is limited by the epigenetic suppression of RIPK3 (Preston et al, 2022; Hoff et al, 2023). Nevertheless, reports of elevated RIPK1, RIPK3, and MLKL expression in progressive liver injury and severe MALFD/MASH suggest that hepatocytes may overcome this defect in certain contexts (Gautheron et al, 2014; Saeed et al, 2019; Afonso et al, 2021; Miyata et al, 2021; Hoff et al, 2023). Alternatively, it has been proposed that RIPK3/MLKL signaling may be active in disease-causing parenchymal cholangiocytes and/or nonparenchymal cells (Gautheron et al, 2014) or that RIPK1 may target MLKL to drive necroptosis during liver disease (Günther et al, 2016; Xu et al, 2019; Majdi et al, 2020). Intriguingly, although we detected RIPK3 in macrophages and infiltrating cells within the livers and VAT of HFD-fed WT and $Mlkl^{-/-}$ mice, it remained silenced in hepatocytes, as well as in hepatic cells chronically exposed to palmitate. Furthermore, we failed to observe phosphorylation of MLKL in the liver and found that neither RIPK3 nor MLKL was essential for hepatic cell death, fitting with reports that hepatocytes largely undergo apoptosis (Akazawa & Nakao, 2018). Consequently, our results support the idea that an inflammatory RIPK3/caspase-8 axis exists in macrophages and argues against necroptotic RIPK3/MLKL signaling in liver cells. Instead, our data suggest that MLKL may have noncanonical activities in obesity and MAFLD.

In our hands, MLKL acts independently of canonical RIPK3 signaling to drive obesity and metabolic dysfunction. How MLKL deficiency protects mice from obesity is unclear, but one recent study proposed that MLKL, and not RIPK3, drives white adipose tissue differentiation (Magusto et al, 2022). MLKL loss also distinctly

---

Data shown are the mean ± SEM, n = 8–12 mice per group pooled from three experiments. Statistical analyses shown were calculated using the median lipid/tissue weight-normalized data after (log$_{10}$) transformation. One-way ANOVA followed by Tukey's multiple comparison test, *$P < 0.05$, **$P < 0.01$, ***$P < 0.001$, ****$P < 0.0001$. **(C, D)** Fold change (log$_2$) of individual lipid species in $Mlkl^{-/-}$ HFD- versus WT HFD-fed (C) serum and (D) liver. Data shown are median-normalized log$_2$-transformed data adjusted for a false discovery rate. Unpaired $t$ test, $P < 0.05$. Key: DG, diglycerides; TG, triglycerides; CE, cholesterol esters; Cer, ceramide; SM, sphingomyelin; AC, acylcarnitine; PC, phosphatidylcholine; PI, phosphatidylinositol; PS, phosphatidylserine; PG, phosphatidylglycerol; PE, phosphatidylethanolamine; COH, cholesterol. Source data are available for this figure.

caused a select reduction in circulating cholesterol, akin to a recent report in a model of atherosclerosis (Rasheed et al, 2020). However, whilst we observed this phenomenon was associated with reduced lipid accumulation in tissues, Rasheed et al observed that MLKL deficiency promotes the retention of lipid in macrophages within the atherosclerotic plaque by directly impairing endocytic trafficking (Rasheed et al, 2020), suggesting differences in MLKL functions between cell types. MLKL has also been acknowledged to have noncanonical actions in the liver to drive MAFLD, including the inhibition of autophagic flux that promotes ER stress, impairment of insulin signaling, and mitochondrial biogenesis, as well as promoting de novo lipogenesis (Saeed et al, 2019; Xu et al, 2019; Majdi et al, 2020; Wu et al, 2020). Our analyses support a dominant role of MLKL in perturbing lipid metabolism in the liver, although we anticipate that these other processes, such as impaired autophagic flux that is RIPK3-independent (Wu et al, 2020), could contribute to lipid accumulation and associated ER stress. Impressively, we observed down-regulation of several key PPARα/γ-induced genes that regulate lipid uptake, transport, synthesis, and storage in the HFD-fed $Mlkl^{-/-}$ liver, with a number of these genes also lowered basally or not induced by palmitate treatment in our MLKL-deficient hepatic cell lines. It remains unclear how MLKL controls the transcription of genes involved in lipid metabolism, as well as membrane receptor signaling, but given MLKL's propensity to target lipid-rich membranes including endosomes, autophagolysosomes, and the nucleus itself, it could be direct or indirect via actions on regulatory processes (Yoon et al, 2016, 2017; Rasheed et al, 2020; Wu et al, 2020; Pradhan et al, 2021). In line with an indirect impact, we detected lower expression of the fatty acid uptake receptor CD36 and transporter FABP4 in the liver and observed less lipid accumulation in $Mlkl^{-/-}$ hepatocytes and hepatic cells. CD36 is well known to signal via PPARs and play a key role in promoting de novo lipogenesis and limiting β-oxidation and autophagy (Wilson et al, 2016; Li et al, 2019). As normal CD36 expression has been reported in atherosclerotic macrophage foam cells lacking MLKL (Rasheed et al, 2020), our work again highlights that MLKL may differentially regulate lipid regulatory molecules and thus affect lipid handling and storage differently depending on the cell type and tissue.

Closely aligning with our transcriptomic data, targeted lipidomics of tissues revealed a shift in the lipidome of MLKL-deficient mice on HFD. Adding to MLKL's new role in adipocyte differentiation (Magusto et al, 2022), we discerned a possible role of MLKL in the synthesis of lipotoxic saturated ceramide species in the VAT, which are associated with reduced adipocyte function and global insulin resistance (Chaurasia et al, 2020). Subsequently, we observed a prominent defect in diglyceride (DG) and triglyceride (TG) production in $Mlkl^{-/-}$ mice, particularly MUFA and PUFA species, showing a new role of MLKL in lipid biosynthesis. Interestingly, in a model of MASH, RIPK3 deficiency has also been associated with a more select reduction in DG and TG species with longer acyl chains and greater double bonds (Afonso et al, 2021), which may suggest some functional overlap between RIPK3 and MLKL in the production of long- and very-long-chain fatty acids that may promote MLKL trafficking and necroptosis (Parisi et al, 2019).

Our study uncovers a role for RIPK3/caspase-8 signaling in regulating obesity-induced inflammation, independent of its capacity to activate NLRP3, aligning with recent reports in sepsis and

arthritis models (Allam et al, 2014; Lawlor et al, 2015). More importantly, we delineate a noncanonical, RIPK3-independent role for MLKL in lipid metabolism and the development of obesity and MAFLD. How MLKL is triggered in this scenario remains elusive. It is still possible that RIPK3 signaling in hepatocytes is below our threshold of detection leading to sublethal necroptosis signaling or that RIPK1 may target MLKL to drive responses during liver disease (Günther et al, 2016; Xu et al, 2019; Majdi et al, 2020). It is also plausible that non-necroptotic MLKL activity may be triggered by an as-yet-unknown event, such as that which has been posited for demyelinating diseases (Ying et al, 2018).

The vital role of MLKL in regulating lipid uptake, transport, and metabolism that we show here suggests future studies investigating the proximity of MLKL to specialized metabolic organelles and examination of the expression and function of lipid receptors/transporters, in the absence of MLKL, are warranted. Ultimately, our study also offers a new avenue and perspective on how targeting divergent MLKL functions, beyond cell death, may limit obesity and MAFLD.

## Materials and Methods

### Study design

The aim of this study was to determine the role of core extrinsic cell death machinery, namely, apoptotic caspase-8 and necroptotic RIPK3 and MLKL in the development of chronic inflammation in obesity and metabolic syndrome. For in vitro experiments, both male and female control and gene knockout mice were used to determine the mode of cell death and pathogenic IL-1β activation triggered by saturated fatty acid palmitate. For in vivo studies, age-matched male control and gene knockout mice were used in normal chow diet (ND) and HFD experiments because of their increased susceptibility to the model. Comparisons were made between the ND- and HFD-fed mice and between genotypes regarding weight gain, as well as glucose and insulin tolerance over 23–25 wk. End-stage organ weights were recorded, and disease pathology was assessed by serological measurements performed by an external commercial service blinded to the groups, liver histopathology was assessed in a blinded fashion by a veterinary pathologist, adipocyte histomorphometry was assessed using an automated imaging script, and inflammation was assessed by flow cytometry. Based on the differences in dyslipidemia, obesity, and MAFLD observed in MLKL-deficient mice, compared with RIPK3-deficient animals, RNA-seq was performed on MLKL-deficient liver tissue and lipidomic analyses on serum, VAT, and liver to assess global changes. The number of samples, combined samples, and independent experiments is included in the figure legends.

### Mice

All mice were housed under standard regulatory conditions at the Walter and Eliza Hall Institute of Medical Research (WEHI), Australia, and Baker Heart and Diabetes Institute, Australia. All procedures were performed in accordance with the National Health and

Medical Research Council Australian Code of Practice for the Care and Use of Animals and approved by the WEHI Animal Ethics Committee or the AMREP AEC. WT, MLKL-deficient ($Mlkl^{-/-}$) (Murphy et al, 2013), RIPK3-deficient ($Ripk3^{-/-}$) (Newton et al, 2004), RIPK3/caspase-8–doubly deficient ($Ripk3^{-/-}Casp8^{-/-}$) (Rickard et al, 2014b), caspase-1–deficient ($Casp1^{-/-}$) (Kuida et al, 1995), and NLRP3-deficient ($Nlrp3^{-/-}$) (Brydges et al, 2009) mice, generated or backcrossed onto the C57BL/6J background, were used for the in vitro generation of BMDMs at > 6 wk of age. For the in vivo high-fat diet model, WT control mice harboring a floxed caspase-8 allele ($Casp8^{lox/lox}$) were first crossed onto a RIPK3-deficient background ($Casp8^{lox/lox}Ripk3^{-/-}$). These mice were then used to generate mice with a conditional deletion of caspase-8 in myeloid cells using the lysozyme M-Cre transgenic mouse ($Casp8^{LysMcre}Ripk3^{-/-}$). To obtain the optimal numbers of age-matched male mice of relevant mouse lines, the following breeding strategies were adopted and genotypes were pooled by postnatal day 35 and acclimatized for at least 3 wk. $Casp8^{lox/lox}$ (or $Casp8^{LysMcre/+}$) mice were generated using $Casp8^{lox/lox}$ x $Casp8^{lox/lox}$ and/or $Casp8^{LysMcre/+}$ x $Casp8^{lox/lox}$ crosses. $Casp8^{lox/lox}Ripk3^{-/-}$ and $Casp8^{LysMcre}Ripk3^{-/-}$ mice were obtained in parallel matings from $Casp8^{lox/lox}Ripk3^{-/-}$ x $Casp8^{LysMcre/+}Ripk3^{-/-}$ crosses. WT and $Mlkl^{-/-}$ mice were generated from heterozygous and/or homozygous $Mlkl^{-/-}$ and WT $Mlkl^{+/+}$ matings. Bone marrow chimeric mice were generated by irradiating C57BL/6 recipient mice twice with doses of 5.5 Gγ, spaced 3 h apart, and intravenously injecting 5 × 10⁶ Ly5.2 WT or $Mlkl^{-/-}$ donor bone marrow cells (post–red blood cell lysis) via the tail vein in 200 μl PBS. Mice were allowed to reconstitute for 8 wk prior to dietary challenge.

## Diets

8- to 9-wk-old WT, $Mlkl^{-/-}$, $Casp8^{lox/lox}$, $Casp8^{LysMcre/+}$, $Casp8^{lox/lox}Ripk3^{-/-}$, and $Casp8^{LysMcre}Ripk3^{-/-}$ mice were fed either a HFD (36% fat, 59% of total energy from lipid; Specialty Feeds) or a normal chow diet (ND) ad libitum for 16–26 wk, as performed previously (Murphy et al, 2016). Cohorts were weighed weekly to measure weight gain. HFD-fed mice (irrespective of the genotype) that did not achieve a 25% weight gain by 25 wk were excluded from further analysis (i.e., deemed non-responders), as were animals that developed malocclusion. In the case of BM chimeras, mouse body composition (lean and fat mass) was measured using a 4-in-1 EchoMRI body composition analyzer (Columbus Instruments). Food input and output were grossly monitored by the amount of food consumed and cage weight on a weekly basis. At specified times, or at the experimental endpoint, blood was collected via cardiac bleed for serum collection by centrifugation. Organs and tissues, including the liver, spleen, kidney, pancreas, SAT, and VAT, were harvested, and weights were recorded before further analysis.

## Glucose and insulin tolerance tests

Intraperitoneal (IP) GTT were performed in ND- and HFD-challenged mice at 8–10 wk and 16–18 wk, and an ITT was performed after 23 wk of diet challenge. In both cases, mice were fasted for 5–6 h before being given an intraperitoneal injection of either 1.5 g D-glucose per kg of body weight or 0.75 U insulin per kg of body weight. Oral glucose tolerance tests (2 g/kg by oral gavage) were performed on bone marrow chimeric mice based on lean body mass. Blood glucose levels were measured (Accu-Chek Performa; Roche) by tail bleeds before injection or oral gavage (time 0 min), and measurements were made at 15, 30, 50, 80 (90), and 120 min post–glucose or insulin administration. For ITT, results were normalized for baseline fasting blood glucose levels and the AUC was analyzed, as previously described (Virtue & Vidal-Puig, 2021).

## Serum analysis

Serum triglyceride, cholesterol, glucose, alanine aminotransferase (ALT), and aspartate aminotransferase (AST) levels were measured by ASAP Laboratory. Insulin and NEFA levels were assessed using a mouse insulin ELISA kit (Promega) and WAKO NEFA-C kit (WAKO), respectively, according to the manufacturer's instructions.

## Histopathology

Liver and VAT biopsies were fixed in 10% (wt/vol) neutral-buffered formalin and paraffin-embedded. Sections (4 μm) were stained with hematoxylin and eosin (H&E), Periodic acid–Schiff, and Sirius Red, or subjected to automated immunohistochemical staining with F4/80 (in-house; WEHI histology services), RIPK3 (in-house; WEHI histology services; 8G7; available from Merck; MABC1595O), or cleaved caspase-3 (Asp 175 CST; 5A1E) antibodies, and signals were detected with a DAB product and sections counterstained with hematoxylin (Chiou et al, 2024). Liver samples were also snap-frozen in Tissue-Tek OCT and 8 μm sections cut for Oil Red O staining to illustrate lipid content. Liver foci of lobular inflammation, steatosis, and hepatocyte ballooning were scored on H&E (or Periodic acid–Schiff)-stained sections, and fibrosis was assessed on Sirius Red–stained section, by an independent pathologist that was blinded to the study details (Gribbles Veterinary Pathology) using the MASH Clinical Research Network criteria.

An automated script was created in FIJI (WEHI Centre for Dynamic Imaging; available from the authors on request) that used a MorphoLibJ plugin to segment adipocytes for quantification of the mean VAT adipocyte size (Schindelin et al, 2012; Legland et al, 2016). Adipocyte measurements were performed in two to four focal regions per H&E-stained tissue section. Areas with < 100 adipocytes within a quadrant (field of view) and quadrants with poor tissue integrity for quantification were excluded from the analysis.

## Flow cytometric analysis

VAT and liver tissue (0.5–1 g) were harvested from mice and minced into small pieces (3–4 mm) with surgical scissors, and then enzymatically dissociated in 1 mg/ml type I collagenase (Worthington) and DNase 1 (5 ng/ml) in 2% (vol/vol) FBS (Bovogen) in DMEM for 45 min at 37°C (vortexing every 5 min) before adding 2.5 mM EDTA for the final 15 min. Cells were sieved through a 70 μm cell strainer (Falcon) and washed in 3% (vol/vol) FBS containing 2.5 mM EDTA in PBS (FACS buffer), and leukocytes were pelleted at 800$g$ for 15–20 min with the brake off. Red blood cells were lysed, and the cells were washed and resuspended in FACS buffer and subsequently stained with fluorochrome-conjugated antibodies from

BioLegend, BD Bioscience, and eBioscience to mouse CD16/32 (Fc block, 2.4G2), CD45.2 (104), CD11b (Mac-1), F4/80 (BM1), Ly6G (1A8), Ly6C (HK1.4), CD11c (N418; in-house), and MHCII (M5/114.15.2; in-house) for 30 min on ice. Cells were washed and resuspended in FACS buffer containing propidium iodide (PI, 1 $\mu$g/ml) and counting beads (123count eBeads; Invitrogen) before analysis on an LSRFortessa instrument using FACSDiva software, and analysis using WEASEL software version 2.7/2.8 (purchased from Frank Battye).

## Preparation of BSA-conjugated palmitate (PA)

To obtain a 5:1 M ratio of palmitate (PA) to BSA, 1% (wt/vol) fatty acid–free BSA (Worthington and Merck) was prepared by dissolving fatty acid–free BSA in serum-free DMEM containing 4 $\mu$M L-glutamine. Stocks of 150 mM PA were prepared by dissolving 41.8 mg sodium palmitate (Sigma-Aldrich) in 1 ml 50% ethanol at 70°C for 5 min. BSA was pre-incubated at 37°C for 30 min before conjugation to PA for 1 h at 37°C.

## BMDM cultures

Bone marrow cells were harvested from the tibial and femoral bones to generate BMDMs. Cells were cultured in DMEM (Gibco) containing 10% (vol/vol) FBS, 15–20% (vol/vol) L929-conditioned media, 4 $\mu$M L-glutamine (Life Technologies), 1 mM sodium pyruvate (Thermo Fisher Scientific), and 100 U/ml penicillin/streptomycin (P/S) (Life Technologies) for 6 d at 37°C, 10% $CO_2$. Unless otherwise indicated, macrophages were plated at $4 \times 10^5$ cells/well in 24-well tissue culture–treated plates (BD Falcon), or at $3 \times 10^5$ in 24-well non-tissue culture–treated plates. Macrophages were primed with B4 or B5 LPS (both at 50 ng/ml; Ultrapure; InvivoGen) or $Pam_3CSK_4$ (500 ng/ml; InvivoGen) for 3 h before the addition of 300–1,200 $\mu$M of fatty acid–free BSA conjugated to palmitate (PA-BSA) or BSA only (to match the maximum BSA added in palmitate stimulations). After ~18 h, cell supernatants were routinely collected from tissue culture for cytokine analysis by ELISA and supernatants and cell lysates collected for immunoblotting. In some cases, cells were harvested from non-tissue culture–treated plates using 5 mM EDTA in PBS. Cell viability was measured by PI (1–2 $\mu$g/ml) uptake, and flow cytometric analysis was performed on a BD LSRFortessa X-20 or BD FACSCanto instrument using FACSDiva software (BD Biosciences). Data were analyzed using FlowJo software version 10.6.1 or WEASEL software.

## Ex vivo VAT cultures

The VAT (0.5 g) was harvested from HFD-challenged mice and cultured in a well of a 24-well tissue culture plate in 0.1% (wt/vol) BSA/DMEM in the presence or absence of LPS (50 ng/ml) for 16–18 h at 37°C, 10% $CO_2$. Supernatants were collected for cytokine analysis.

## Cytokine analysis

IL-1$\beta$ (R&D) and TNF (eBioscience) ELISA kits were used, according to the manufacturer's instructions. For detection of TNF in BMDM supernatants, samples were diluted 1:10 in an assay diluent.

## Cell line generation, CRISPR/Cas9 gene editing, and expression systems

MLKL-deficient murine Hepa1-6 hepatic cells were generated based on a CRISPR/Cas9 protocol described previously (Baker & Masters, 2018). pFU-Cas9-mCherry plasmid DNA (provided by Marco Herold, WEHI; available from Addgene) was transiently transfected into HEK293T cells alongside pMDLg (packaging; Addgene), RSV-REV (packaging; Addgene), and VSVg (envelope; Addgene) using FuGENE (Promega) diluted in Opti-MEM (Thermo Fisher Scientific) to generate lentiviral particles in DMEM supplemented with 10% (vol/vol) FBS, 50 $\mu$g/ml penicillin G, 50 U/ml streptomycin, 1 mM sodium pyruvate, and 2 mM L-glutamine (Gibco). The cell culture supernatant was collected 48 h later and filtered through a 0.45 $\mu$m filter before cell transduction. Lentiviral transduction was performed by replacing normal cell culture medium with DMEM containing lentivirus particles and 10 $\mu$g/ml polybrene for 24 h. Transduced, Cas9-mCherry–positive cells were selected via FACS of mCherry+ cells. To generate MLKL-deficient cells, lentiviral particles harboring *Mlkl* targeting (exon 4 5' gtcttcagtttggtccacgg) sgRNAs were cloned into the pkLV-U6sgRNA-EF(BbsI)-PGKpuro2ABFP plasmid transduced into Cas9+ Hepa1-6 cells using the method described above. Transduced cells were selected by FACS of mCherry+/BFP+ cells, then diluted and plated as single cells to obtain clonal populations (termed *Mlkl$^{-/-}$* clone #1 and #3). Gene disruption was confirmed by sequencing, immunoblot analysis of the targeted protein, and functional analysis.

RIPK3-WT was synthesized by ATUM and cloned into an N-terminal FLAG-tagged DOX-inducible, puromycin-selectable plasmid (pF TRE3G PGK), as previously described (Moujalled et al, 2013; Murphy et al, 2013; Frank et al, 2022). RIPK3 transgene–containing lentivirus was generated, as described above, and WT (Cas9+) and *Mlkl$^{-/-}$* Hepa1-6 cells were infected in media containing 10 $\mu$g/ml polybrene. RIPK3 transgenic cells were selected in 5 $\mu$g/ml puromycin (Gibco) for 2 d. For transgene validation, WT and *Mlkl$^{-/-}$* Hepa1-6 cells expressing the DOX-inducible RIPK3 were lifted using TrypLE (Sigma-Aldrich), plated out at $4 \times 10^5$ cells per well in a 24-well tissue culture plate, and rested overnight at 5% $CO_2$, 37°C. RIPK3 expression was then induced with 100 ng/ml DOX for at least 3 h, and cells were treated, as indicated in the figure legends, with recombinant human TNF (100 ng/ml; R&D) and birinapant (1 $\mu$M; kindly provided by TetraLogic Pharmaceuticals) alone or in combination with Q-VD-OPh (40 $\mu$M; MedChemExpress) for 12–16 h. RIPK3 induction efficiency was confirmed by immunoblot analysis, as detailed below, and functional effects on cell death were measured by flow cytometric analysis of PI (2 $\mu$g/ml) uptake on an LSRFortessa instrument using FACSDiva software. For palmitate stimulations, WT, *Mlk1$^{-/-}$* #1, and *Mlk1$^{-/-}$* #3 Hepa1-6 reconstituted with DOX-inducible RIPK3-WT were plated at $4 \times 10^5$ cells per well in a 24-well tissue culture plate (immunoblot, cell death, BODIPY staining) or $8 \times 10^5$ cells per well in a 12-well tissue culture plate (qRT–PCR) and rested overnight. Cells were stimulated with or without DOX (100 ng/ml) for 3 h and media replaced with fresh media immediately before the addition of fatty acid–free BSA (Merck) or 100–300 $\mu$M PA-BSA. Cells were analyzed for appropriate readouts.

## BODIPY staining

Hepa1-6 cells were cultured in the presence or absence of fatty acid–free BSA (Merck) or 100 or 200 $\mu$M sodium palmitate (Sigma-Aldrich) conjugated to fatty acid–free BSA at 37°C, 5% $CO_2$ for 24 h. The supernatant was aspirated, and cells were washed with PBS, then incubated with 2 $\mu$M BODIPY 493/502 (Cayman Chemical) in PBS for 15 min at 37°C, 5% $CO_2$. The supernatant was aspirated, and cells were washed with PBS, then harvested with TrypLE (Gibco), resuspended in PBS + 2% fatty acid––free BSA + 5 mM EDTA, and acquired on a Fortessa X-20 (BD Biosciences). Relative BODIPY was calculated by dividing the geometric mean fluorescence intensity (determined using FlowJo software; BD Biosciences) of each sample by the average geometric mean fluorescence intensity of the WT untreated group.

## Immunoblotting

Cell lysates and supernatants were boiled for 10 min in 1× NuPAGE LDS (Thermo Fisher Scientific) or in-house (2% [wt/vol] SDS, 10% [vol/vol] glycerol, 50 mM Tris, pH 6.8, 0.01% bromophenol blue) sample buffer containing 5% (vol/vol) $\beta$-mercaptoethanol ($\beta$-ME). Samples were separated on 4–12% Bis-Tris gradient gels (NW04125/27BOX; Invitrogen), and proteins were transferred onto a nitro-cellulose membrane (Millipore). Ponceau staining was used to confirm protein transfer and as a loading control. Membranes were blocked with 5% (wt/vol) skim milk in TBS containing 0.1% (vol/vol) Tween-20 (TBS-T) for 1 h and then probed overnight at 4°C with the following primary antibodies (all diluted 1:1,000 in 5% [wt/vol] skim milk in TBS-T containing 0.02% sodium azide, with the exception of $\beta$-actin that was diluted 1:5,000): pro- and cleaved IL-1$\beta$ (AF-401-NA; R&D), pro- and cleaved caspase-1 (AG-20B-0042-C100; Adipogen), pro-caspase-8 (3B10; WEHI), cleaved caspase-8 (9429S; CST), NLRP3 (AG-20B-00140-C100; Adipogen), RIPK3 (WEHI; 1H12 or WEHI, 8G7; available from Merck; MABC1595) (Murphy et al, 2013; Petrie et al, 2019), MLKL (WEHI; 3H1; available from Merck; MABC604) (Samson et al, 2021), pMLKLS345 (196436; Abcam), GAPDH (D4C6R; CST), and horseradish peroxidase–conjugated $\beta$-actin (5125S and 13E5; CST). Relevant HRP-conjugated primary and secondary antibodies (all diluted 1:5,000) applied for 1 h at RT in 5% (wt/vol) skim milk in TBS-T. Membranes were washed 6x in TBS-T between each incubation. Membranes were developed using the Immobilon Forte Western HRP substrate (WBLUF0500; Merck Millipore) and imaged with Bio-Rad ChemiDoc MP or Invitrogen iBright Imaging System. Images were analyzed and processed with Bio-Rad ImageLab or iBright analysis software.

The snap-frozen liver (50–100 mg) tissue was ground using a mortar and pestle on dry ice, and tissue was homogenized (by pipetting) in 300–600 $\mu$l RIPA buffer (150 mM NaCl, 50 mM Tris [pH 7.4], 1 mM deoxycholate, 1% [vol/vol] Triton X) containing cOmplete protease inhibitor cocktail (Roche) and PhosSTOP (Roche) and agitated on a rotating wheel for 1 h at 4°C. After centrifugation (20,000$g$) at 4°C for 15 min, the lipid layer was discarded, and the supernatant was transferred to a fresh tube. This process was repeated two more times. The tissue lysate protein concentration was quantified using the DC protein assay (Bio-Rad) or BCA assay (Pierce) according to the manufacturer's instructions, and 40 $\mu$g of

tissue lysate was analyzed by immunoblotting, as above, with antibodies against CD36 (affinity-purified polyclonal antibody SR-B3; R&D Systems, In vitro Life Science), FABP4 (2120; CST), RIPK3 (WEHI; 1H12 or WEHI; 8G7; available from Merck; MABC1595), MLKL (WEHI; 3H1; available from Merck; MABC604), phospho-MLKL S345 (196436; Abcam), phospho-RIPK3 T231/S232 (a gift from Genentech; GEN135-35-9), and $\beta$-actin, and relevant secondaries. In some cases, densitometry was performed using iBright or ImageLab software and relevant proteins were normalized to $\beta$-actin and then expressed as a fold change over WT ND–fed tissue.

## Quantitative RT–PCR

Liver biopsies were snap-frozen, and 50 mg of tissue, were ground in a mortar and pestle over dry ice. RNA from the tissue was extracted using TRIzol RNA Isolation Reagents (Life Technologies) with DNase treatment and purification performed using the ISO-LATE II RNA mini kit (Bioline) or RNeasy mini kit (QIAGEN) according to the manufacturer's instructions. RNA from Hepa1-6 cells ($8 \times 10^5$) was extracted using the RNeasy mini kit with on-column DNase treatment using RNase-free DNase set (QIAGEN), according to the manufacturer's instructions. RNA concentration and purity were quantified using NanoDrop 2000 Spectrophotometer, and cDNA synthesis was performed using a High-Capacity cDNA synthesis kit (Applied Biosystems, Thermo Fisher Scientific). qRT-PCR was then performed using Power SYBR Green PCR Master Mix (Applied Biosystems) on a QuantStudio 6 Flex PCR system (Thermo Fisher Scientific) with the primer pairs listed in Table S1. Relative mRNA levels were calculated using the comparative delta–delta Ct ($\Delta\Delta$Ct) method ($2^{-[(\Delta Ct\ genotype/diet) - (\Delta Ct\ WT\ ND)]}$) or ($2^{-[(\Delta Ct\ genotype/stimuli) - (\Delta Ct\ WT\ BSA)]}$), where $\Delta$Ct values were obtained by normalization to the internal housekeeping reference gene $18s$. Specificity of each primer set was confirmed by the observation of a single peak in the melt curve graph of each qRT-PCR run.

## 3′ mRNA sequencing

RNA from liver samples was extracted as above. Integrity of RNA was examined using TapeStation Agilent 4200, and samples with RIN>8 were selected for library preparation for 3′ mRNA-sequencing analysis. 3′ mRNA-sequencing libraries were prepared using 100 ng of total RNA using QuantSeq 3′ mRNA-seq Library Prep (Lexogen) according to the manufacturer's instructions. The single-end 75-bp reads were demultiplexed using CASAVAv1.8.2, and Cutadapt (v1.9) was used for read trimming (Martin, 2011). The trimmed reads were subsequently mapped to the mouse genome (mm10) using HISAT2 (Kim et al, 2019). FeatureCounts from the Rsubread package (version 1.34.7) was used for read counting after which genes without a counts per million reads (CPM) in at least three samples were excluded from downstream analysis (Liao et al, 2014, 2019). Count data were normalized using the trimmed mean of M-values (TMM) method, and differential gene expression analysis was performed using the limma-voom pipeline (limma, version 3.40.6) (Robinson & Oshlack, 2010; Law et al, 2014; Liao et al, 2014). Comparisons between WT ND, $Mlkl^{-/-}$ ND, WT HFD, and $Mlkl^{-/-}$ HFD were made. GSEA 2.2.2 was used for GSEA (Subramanian et al, 2005;

Liberzon et al, 2015). Gene ontology (GO) analysis was performed using *Metascape*, and *pheatmap* (version 1.0.12) was used to generate heatmaps. The datasets generated during this study are available at GEO225560.

### Lipidomic analysis

Targeted lipidomic analysis using liquid chromatography–mass spectrometry (LC-MS) was performed on the serum, liver, and VAT of ND- and HFD-fed WT and $Mlkl^{-/-}$ mice. Briefly, for serum, 20 $\mu$l was aliquoted (serum from up to three mice was pooled where necessary) into Eppendorf tubes and 380 $\mu$l of cold 2:1 chloroform/methanol containing four internal standards (10 mg/l; PG 17:0/17:0, PC 19:0/19:0, PE-D31 16:0/18:1, and TG-D5 19:0/12:0/19:0) (Product # 860374, 850367, 8609040, and 830456 from Avanti Polar lipids) was added. For liver and VAT lipid extraction, the snap-frozen tissue (~30 mg) was transferred into cryomill tubes and 500 $\mu$l of cold 1:9 chloroform: methanol (vol/vol) containing four internal standards as mentioned earlier (10 mg/l) was added before homogenization using a cryogenically cooled bead-mill (6,800 rpm, 3 × 45 s cycles; Precellys 24 coupled to Cryolys unit from Bertin Technologies). Next, 400 $\mu$l of homogenate was transferred into Eppendorf tubes and 680 $\mu$l of 100% methanol was added to make a final concentration of 2:1 chloroform: methanol (vol/vol). Serum, VAT, and liver samples were then vortexed (30 s), mixed with a thermomixer (Eppendorf South Pacific Pty Ltd) at 950 rpm for 10 min at 20°C, and centrifuged at 15,000 rpm (Beckman Coulter Microfuge 22R refrigerated microcentrifuge; Beckman Coulter Australia Pty Ltd) for 10 min at room temperature. 1250 $\mu$l of VAT and liver or 350 $\mu$l of serum supernatant were then transferred into Eppendorf tubes containing glass inserts and gradually evaporated (50 $\mu$l at a time) using a vacuum concentrator, with a temperature maintained at 30–35°C (Christ RVC 2-33; Martin Christ Gefriertrocknungsanlagen GmbH). Samples were reconstituted with 10 $\mu$l methanol: 90 $\mu$l water-saturated butanol (100 $\mu$l, vol/vol) for LC-MS analysis. Pooled biological quality controls were prepared by pooling aliquots of the extracts from each sample and were run after every five samples.

Extracted lipids were processed and detected by Metabolomics Australia (Bio21 Institute, Melbourne, Australia) using an Agilent 1290 LC system and Agilent Triple Quadrupole 6490 mass spectrometer (Agilent Technologies Australia), as previously described (Huynh et al, 2018). Briefly, lipids from the serum, VAT, and liver samples were separated using a Zorbax Eclipse Plus C18 column (100 mm × 2.1 mm × 1.8 $\mu$m; Agilent Technologies Australia). Injection volume was kept at 1 $\mu$l with a LC flow rate of 400 $\mu$l/min. The LC mobile phase solvents were acetonitrile/water/isopropanol (30:50: 20, vol/vol/vol) for mobile phase A and acetonitrile/water/isopropanol (9:1:90, vol/vol/vol) for mobile phase B with 10 mM ammonium formate for both A and B. The LC gradient, MS parameters, and the targeted dynamic scheduled multiple reaction monitoring (dMRM) transitions of each lipid species have been previously described (Huynh et al, 2018, 2019). Quantitation was based on relative changes in peak areas. Data processing was performed using Agilent's Mass Hunter Quantitative Analysis software (Agilent Technologies Australia). Lipids were named

according to the nomenclature described in LIPID MAPS (Liebisch et al, 2013).

For analysis, raw data were normalized for median lipid content per mouse and the weight of tissue was analyzed, as appropriate. These data were subjected to a $\log_{10}$ transformation for statistical analyses. Total relative abundance of lipid species classes was calculated by summation of individual normalized species, before $\log_{10}$ transformation for statistical analyses (mean ± SEM). The $\log_2$ fold change was also calculated in MetaboAnalyst 5.0 from the median- and weight-normalized HFD sample data and adjusted for a false discovery rate to detect significantly different lipid species in the serum and liver ($P < 0.05$, $t$ test). Heatmaps were generated using GraphPad Prism (version 9) software and are presented as median abundance of lipid species. The raw data and internal quality control samples are available on request.

### Statistical analysis

All graphical data are presented as the mean ± SEM for biological samples or SD for replicates, as indicated in the figure legends. The AUC was calculated for weights, % weight gain, and GTT and ITT. Statistical comparisons between two genotypes were performed using a $t$ test and between three genotypes, a one-way analysis of variance (ANOVA) was performed with a Tukey or Dunnett's post hoc correction for multiple comparisons. A one-way ANOVA with a Tukey post hoc correction for multiple comparisons was used for comparisons between treatments and multiple genotypes. Analyses were performed using GraphPad Prism (version 9) software, and a $P$-value < 0.05 was considered statistically significant.

## Data Availability

RNA-seq data were uploaded to the Gene Expression Omnibus repository, and the accession number is GSE225560. Lipidomic data are available upon request. All data needed to evaluate the conclusions in the article are present in the article, supplementary material, and source data.

## Supplementary Information

## Acknowledgements

We thank WEHI Bioservices and Alfred Research Alliance Animal Facility for expert animal care, and WEHI's histology laboratory and the Centre for Dynamic Imaging for histopathology staining and microscopy assistance, respectively. We thank WEHI and Monash FlowCore for assistance with FACS. We thank Professor John Silke, Professor Terry Speed (WEHI), Professor Brian Drew (Baker Heart and Diabetes Institute), and Professor Richard Ferrero for reagents and helpful advice. We gratefully acknowledge grant support from the National Health and Medical Research Council (NHMRC) of Australia: project grants (1140187, 1165591 to ED Hawkins; 1145788 to JE Vince, KE Lawlor, and JM Murphy; 1101405 to JE Vince; 1162765 to KE Lawlor), Ideas grants

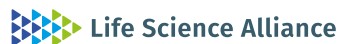

(2011584 to M Speir and JM Hildebrand; 1183070 to JE Vince; 1181089 to KE Lawlor), and fellowships (CJ Martin Overseas Biomedical Training Fellowship 1144014 to SA Conos; CDF/Leadership Fellowships 1172929 to JM Murphy; 2008652 to ED Hawkins; 1194329 to AJ Murphy; 1141466, 2008692 to JE Vince). KE Lawlor is Australian Research Council (ARC) Future Fellow (FT190100266). This work was also supported by operational infrastructure grants through the Australian Government Independent Research Institute Infrastructure Support Scheme (9000719) and the Victorian State Government Operational Infrastructure Support, Australia.

## Author Contributions

H Tye: data curation, formal analysis, investigation, methodology, and writing—original draft, review, and editing.

SA Conos: data curation, formal analysis, investigation, methodology, and writing—original draft, review, and editing.

TM Djajawi: data curation, formal analysis, investigation, methodology, and writing—original draft, review, and editing.

TA Gottschalk: data curation, formal analysis, investigation, methodology, and writing—review and editing.

N Abdoulkader: data curation, formal analysis, investigation, and methodology.

IY Kong: data curation, formal analysis, investigation, methodology, and writing—original draft, review, and editing.

HL Kammoun: data curation, formal analysis, investigation, methodology, and writing—review and editing.

VK Narayana: formal analysis, investigation, methodology, and writing—review and editing.

T Kratina: formal analysis and investigation.

M Speir: funding acquisition, investigation, and writing—review and editing.

J Emery: investigation.

DS Simpson: investigation and writing—review and editing.

C Hall: investigation.

AJ Vince: investigation.

S Russo: investigation.

R Crawley: investigation.

M Rashidi: investigation.

JM Hildebrand: resources and funding acquisition.

JM Murphy: resources and funding acquisition.

L Whitehead: formal analysis, investigation, methodology, and writing—original draft.

DP De Souza: resources, formal analysis, supervision, investigation, methodology, and writing—review and editing.

SL Masters: resources and writing—review and editing.

AL Samson: data curation, formal analysis, investigation, methodology, and writing—review and editing.

N Lalaoui: resources, supervision, and writing—review and editing.

ED Hawkins: resources, supervision, and funding acquisition.

AJ Murphy: data curation, formal analysis, supervision, investigation, methodology, and writing—review and editing.

JE Vince: conceptualization, formal analysis, supervision, funding acquisition, investigation, methodology, project administration, and writing—original draft, review, and editing.

KE Lawlor: conceptualization, resources, data curation, formal analysis, supervision, funding acquisition, investigation, methodology, project administration, and writing—original draft, review, and editing.

## Conflict of Interest Statement

JM Hildebrand, JM Murphy, and AL Samson contribute to, and KE Lawlor has consulted for, a project developing necroptosis inhibitors with Anaxis Pharma Pty Ltd. All other authors have no competing interests to declare.

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
