## [Reviewer comments · Life Science Alliance]

Life Science Alliance

Divergent roles for RIPK3 and MLKL in high-fat diet induced obesity and MAFLD in mice

Hazel Tye, Stephanie Conos, Tirta Djajawi, Timothy Gottschalk, Nasteho Abdoukader, Isabella Kong, Helene Kammoun, Vinod Narayana, Tobias Kratina, Mary Speir, Jack Emery, Daniel Simpson, Cathrine Hall, Angelina Vince, Sophia Russo, Rhiannan Crawley, Maryam Rashidi, Joanne Hildebrand, James Murphy, Lachlan Whitehead, David De Souza, Seth Masters, Andre Samson, Najoua Lalaoui, Edwin Hawkins, Andrew Murphy, James Vince, and Kate Lawlor

DOI: <https://doi.org/10.26508/lsa.202302446>

Corresponding author(s): *Kate Lawlor, Hudson Institute of Medical Research*

Review Timeline:

Submission Date:	2023-10-18
Editorial Decision:	2023-11-29
Revision Received:	2024-09-26
Editorial Decision:	2024-10-17
Revision Received:	2024-10-30
Accepted:	2024-10-30

Transaction Report:

November 29, 2023

Re: Life Science Alliance manuscript #LSA-2023-02446-T

Dr. Kate E Lawlor
Hudson Institute of Medical Research
Centre for Innate Immunity and Infectious Diseases
27-31 Wright Street
Clayton, Vic 3165
Australia

Dear Dr. Lawlor,

Thank you for submitting your manuscript entitled "Divergent roles for caspase-8 and MLKL in high-fat diet induced obesity and NAFLD in mice" to Life Science Alliance. The manuscript was assessed by expert reviewers, whose comments are appended to this letter. We invite you to submit a revised manuscript addressing the Reviewer comments.

Thank you for this interesting contribution to Life Science Alliance. We are looking forward to receiving your revised manuscript.

Sincerely,

B. MANUSCRIPT ORGANIZATION AND FORMATTING:

Reviewer #1 (Comments to the Authors (Required)):

The overall message of this article is intriguing and clearly highlighting an interesting scientific novelty which includes MLKL role in obesity independently of RIPK3 seemingly via transcriptional upregulation or/ and potentially via regulating metabolic organelles and lipid receptors. While I do think that this, needs further future investigations, I also believe that both the cell death field and the metabolism field would benefit from this piece of evidence being published in this journal.

The manuscript takes almost a Pindaric inspiration, starting with the RIPK3-caspase-8 role in signaling independent of its function to activate NLRP3, then going through the role of Casp8 in obesity which has been largely published and then ending with the role of MLKL in obesity and which is the novel part of the story.

While I think that, it is nice that the authors gave such a good set up to their study, I do find that the first half of the manuscript is something that was quite known already. It is true that there are many clarifications in their approach however most of this could be included in supplementary material. This is because the real interest comes with MLKL and in the first part the authors never include MLKL into their story or systems, cellular or animal.

The MLKL dependent role in obesity is very intriguing and opens many questions which are highlighted by the authors.

Major points:

1)The authors should at least attempt to utilize a cellular system to support some of their speculation with regards to the way that MLKL might mediate this transcriptional regulation.

Does it translocate to the nucleus? Is it associated to a transcription factor?

2)The authors also comment on the fact that MLKL is found bound to lipids and cite, among many, Quarato et al. from the Green lab. However, it is important to specify that such ability of MLKL to bind the lipids in the membrane or lipids in general, necessitates the initial RIPK3 phosphorylation which allows the conformational change needed for the 4HBD to form oligomeric structures. It is really unclear to me how the authors envisage the ability of MLKL to bind lipids independently of RIPK3 or independently of the only known phosphorylation event capable of driving its open conformation. On this point it would be nice if the authors could determine if MLKL localizes in liver adipocytes or around them or from extracts? What about RIPK3?

3) Additionally, MLKL has been shown to be ubiquitinated on different sites leading also to different outcomes: degradation (Silke lab), cell death (Meier Lab) and autophagy (Wallack lab). Considering the autophagy angle and also the role in exosome release (also from the Wallack lab) I wonder if the authors could provide any evidence of MLKL ubiquitination and/or autophagy activation that would justify the defective uptake of fatty acids. This might still remain correlative, however would provide support to some of the conclusions that have been discussed

4) I am not convinced that the data provided can completely exclude the dependency from RIPK3. This is because neither the MLKL/RIPK3 DKO nor the RIPK3 KO alone has been employed to show the differences between these two animal models at least in terms of transcription. Additionally, it would be interesting to compare the Casp8LysMcre/RIPK3^{-/-} with the Casp8LysMcre / Mlkl^{-/-}. According to the authors conclusions the distinct roles of Casp8 and MLKL should then provide a further advantage? I completely understand that this might be a rather long experiment that would involve several animal crosses, hence perhaps the utilization of caspase inhibitor in the background of MLKL deficiency could be an option.

Reviewer #2 (Comments to the Authors (Required)):

In the manuscript by Tye et al., the authors investigated the roles of key cell death players in high fat diet-induced inflammation and lipid metabolism using genetically modified mice. The manuscript appears to be composed of two distinct sections. In the first part, the authors reported that the absence of both Casp8 and Ripk3 conferred resistance to LPS/palmitic acid-induced inflammation in vitro and high fat diet-induced steatosis in vivo. In the second part, the authors showed that the absence of Mlkl also reduced lipid accumulation through the transcriptional regulation of lipid metabolism. Although experiments are all conducted in a scientifically sound manner, several critical data are missing. Moreover, the phenotypes of Mlkl-deficient mice induced by high fat diet was previously published (Saeed et al., PMID: 31132314, DOI: 10.1111/jgh.14740), and more in-depth analysis of molecular mechanisms may be necessary. Therefore, I would suggest that following points should be addressed to improve the scientific quality of this manuscript.

MAJOR POINTS

1. Although Casp8, Ripk3, and Mkl1 are all critical proteins of cell death machinery, the evaluation of cell death is only presented for BMDMs in vitro (PI uptake), but not for genetically modified mice in vivo. I think that Figs 3 and 5 should also contain the photographs and quantitative data of cell death. TUNEL staining may detect various modes of cell death (apoptosis, necroptosis, or pyroptosis). Staining of cleaved caspase-3 may be useful to evaluate whether apoptosis still occurs in the absence of Casp8 and Ripk3. Also, as liver fibrosis was evaluated by Sirius red staining, representative images of Sirius red photographs may be necessary for Figs 3 and 5.

2. In Figures 1F and S1J, it is formally necessary to include the Western blots of MLKL and RIPK3 to indicate the absence of these proteins in BMDMs. If possible, lack of these proteins should be confirmed in the Western blots of fat and liver tissues.

3. In the GO analysis of Figure 6C, the authors focused on the downregulation of fatty acid metabolism in Mkl1 KO mice fed normal diet. However, it appears equally important that cholesterol biosynthesis is upregulated in Mkl1 KO mice under normal diet. These results suggest that both fatty acid and cholesterol metabolic pathways are altered in the absence of MLKL. Although the amounts of serum and liver cholesterol contents shown in Fig 7 may not reflect the transcriptional upregulation of cholesterol biosynthesis, it may also be insightful to examine the expression of key genes in cholesterol metabolism, such as SREBP2, HMG-CoA synthase, or HMG-CoA reductase.

MINOR POINTS

1. It has been reported that extracellular addition of palmitic acid induces ER stress. Please discuss the potential contribution of ER stress in this experimental system.

2. In the titles of GO analysis graphs in Figs 6 and S4, "FC" should be "logFC".

3. In Line 117, reference of Fig S1J should be moved to Line 146.

4. In Lines 145-6, the authors described that "appreciable IL-1 β (p17) secretion in Ripk3 $^{-/-}$ Casp8 $^{-/-}$ BMDMs". However, IL-1 β in supernatant is very low in BMDMs derived from Ripk3 $^{-/-}$ Casp8 $^{-/-}$ mice compared to those from mice of the indicated genotypes (Fig S1J). Is this interpretation correct?

5. In Line 146, reference may be "Fig. S1I to J".

6. In Line 149, reference may be "Fig 1F and H and S1K and L".

7. The reference of Fig S2O (gating strategy of liver FACS) is missing. They may be in Line 207 and Line 246. Fig S2N (gating strategy of VAT FACS) may also be referenced in Line 240.

8. In Line 284, the reference may be "Fig S4C and E and Fig S5a" for HFD-fed mice.

9. In Line 768, "J.H.M" may be "J.M.H" (Joanne M Hildebrand?).

Reviewer #3 (Comments to the Authors (Required)):

Authors provide evidence of the role of Casp8-Ripk3 axes in macrophages in the development of NAFLD in the context of obesity and a non canonical role of MLKL independent of its classical cell death role. The study is relevant to the field and its publication increases the evidence of the role of cell death components in metabolism.

Authors uncovered that RIPK3-caspase-8 signaling is not mandatory for NLRP3 inflammasome activation in LPS treated macrophages exposed to palmitate, but instead caspase-8 regulates inflammatory gene transcription, direct IL-1 β proteolysis, and to a lesser extent cell death in LPS and palmitate treated macrophages. In addition, their in-vivo flow cytometry data reveal less recruitment of inflammatory immune populations to the liver of obese mice, improvement of several metabolic parameters and reduce steatosis.

On the other hand, the manuscript shows a new role of MLKL in lipid metabolism that seems to be independent of its cell death functions. Data shows that MLKL deletion prevents liver steatosis, reduces adipocyte size and gain weight, while at the same time reduces systemic inflammation independently of immune cell activation of MLKL. Revealing the vital role for MLKL in regulating lipid uptake, transport and metabolism. Authors also mention the need for future studies investigating the proximity of MLKL to specialized metabolic organelles and examination of the expression and function of lipid receptors/transporters.

Main general concern

Throughout the first figures it is evident that the protection from obesity-induced metabolic dysfunction is afforded by RIPK3 deficiency rather than by the deficiency of Casp8 in myeloid cells. Therefore, the conclusion of their data in Figures 2 and 3 needs dramatic revision (including the title). If the authors cannot look into metabolic features in Casp8 in myeloid KO mice because of the leaky model, then it is very difficult to draw any conclusion on the Casp8 angle given the data provided. This also means that the findings here are in line with the previous findings by Gautheron et al. Since the evidence on the role of RIPK3 and MLKL in obesity-induced metabolic dysfunction is indeed conflicting, this side-by-side comparison between RIPK3 and

MLKL deficiencies is valuable, despite the fact that there is no novelty on the phenotype of RIPK3 knockout mice upon HFD. Another concern is the visible lack of connection between Figure 1 and the rest of the paper. If the authors see a clear connection this should be better conveyed in the text and perhaps some experimental evidence of this connection should be provided.

Specific comments

- Please clarify what is meant here:

157 and caspase-8 deletion is inefficient

158 upon myeloid-specific deletion (Lysozyme M Cre transgene) (Kang et al., 2004), we chose to

159 compare mice lacking caspase-8 conditionally in myeloid cells on a RIPK3-deficient background

160 (Casp8LysMcreRipk3^{-/-}) (Vince et al., 2018) that will have a more robust loss of RIPK3-caspase-8

161 signaling in myeloid cells compared to RIPK3 deficiency alone.

Are you comparing with Casp8Lysmye alone? Or with RIPK3 KO? If the latter, then this phrase is superficial since in this case Casp8 is wt...

- Fig 1 legend: (C) what are the technical replicates for the authors? Are each of the dots BMDMs isolated from 4 or 5 different mice or just the same cells treated in different wells? If they are not individual mice then the representation is wrong, technical replicates need to be expressed as average, and the average is the result of one individual experiment. On the contrary, each mouse can be considered as an individual experiment, so if this are cells isolated from 4 different mice (and kept separated), then these are individual experiments.

- Fig 1 E: ANOVA analysis. Please explain the question that you wanted to answer when you compared the MLKO KO against RIPK3/casp8 KO in the Palmitic acid treatment. Which questions did you have behind the design of this test? If this is a 2-way ANOVA, which are the 2 parameters that you are comparing? Genotype among the same treatment is 1-way (and you can select the comparisons pairs according to your experimental design), Same genotype, two different treatments, is also one comparison. For (G) the statistical test seems to be different.

- Fig 2 D. the authors mention that RIPK3 KO mice tend to be more obese (line 172). This is not shown in any of the figures in the paper, including supplementary. The meaning of this conclusion should be better explained or revisited.

- Please clarify the following

176 No overall difference in weight gain or organ mass was observed in genetic control mice

177 lacking caspase-8 only in myeloid cells (Casp8LysMcre) that display inefficient deletion (fig. S2G

178 and H) (Kang et al., 2004; Lawlor et al., 2015), suggesting that removal of upstream RIP kinase

179 machinery is required to see the full impact of apoptotic caspase-8 activity.

Do you mean that upstream RIP kinase machinery is required to see the full impact of the loss of caspase 8 activity? So, the prevention of apoptosis? In any case, as mentioned in the general statement, it seems that RIPK3 deficiency is playing a major role in your assessment.

- Comment: explain why Casp8LysMcreRipk3^{-/-} mice can be better in glucose clearance but no differences are showed in insulin tolerance or leave it as an open question. This observation might be extremely important to understand the phenotype, since in general rules, mice fed with HFD become first insulin resistant and latter, glucose intolerant.

In addition, in supplementary figure 2, the ITT is expressed as % relative to initial glucose. This way of quantification could lead to misinterpretation. I suggest using AoC (Area over the curve) suggested by Virtue and Vidal-Puig, Nat Met 2021 to be certain that there are no differences between the groups.

- It is surprising that the conclusion is that MLKL does not contribute to inflammation given that there is a substantial decrease in the proportion of monocyte/macrophages in the liver (Figure 5). Can the authors clarify or revise this conclusion?

- The authors conclude that (line 255) "...reduced adipocyte hypertrophy and liver damage observed in aged or HFD-fed MLKL-deficient mice, suggests that MLKL activity alters tissue homeostasis to cause obesity-induced metabolic disease...." This conclusion arises from experiments in which they transplant either WT or MLKL KO BM into WT recipients. This conclusion seems to be too broad and not reflecting a focused analysis of their data. In reality, their data shows that MLKL (not sure you can say anything about its "activity") in non haematopoietic cells alters tissue homeostasis to cause obesity-induced metabolic disease.

Regarding referee cross-comments, perhaps further in vivo experiments requested by reviewer 1 are too ambitious or too challenging. However, I agree that one cannot exclude a role for RIPK3 in MLKL KO mice and this could be discussed.

We thank the editor and reviewers for their constructive comments. We have carefully revised our manuscript **LSA-2023-02446-T** now entitled “**Divergent roles for RIPK3 and MLKL in high-fat diet induced obesity and NAFLD in mice**” for clarity of language, updated figures and text to match with the *Life Science Alliance* guidelines. We also provide a point-by-point rebuttal addressing all the reviewer comments/questions shown in italics below. Changes to the text, including restructuring, are underlined in the manuscript with updated references.

It is worth highlighting that by performing the requested experiments we have significantly strengthened our findings and, accordingly, revised our discussion points to reflect this. Major points that are made in our revised manuscript are that:

1. We show that, despite the growing number of diseases where NLRP3 acts as a sensor of apoptotic caspase-8- or necroptotic MLKL-mediated cell death signaling, neither are obligatory for NLRP3 inflammasome activity in macrophages exposed to saturated fatty acids. Instead, we show that RIPK3-caspase-8 signaling dominantly regulates the transcription of inflammasome machinery and pro-inflammatory cytokines but can also partly contribute to cell death and direct activation of IL-1 β .
2. We identify that RIPK3 signaling via caspase-8 in myeloid cells drives inflammation, particularly in the visceral adipose tissue, to promote progression to NAFLD.
3. We discover that MLKL, but not upstream kinase RIPK3, contributes to obesity development with ageing and upon high fat diet feeding. Importantly, this divergence in phenotype is largely unexpected as RIPK3 is dominantly thought to couple with MLKL to drive necroptotic cell death responses.
4. We now show that *in vivo* and *in vitro* that hepatocytes are not competent to undergo canonical RIPK3-MLKL signaling to saturated fatty acids.
5. By generating transcriptomic and lipidomic datasets, we revealed that MLKL may noncanonically promote NAFLD by inducing lipid metabolism and perturbing fatty acid synthesis, which represents a breakthrough in our understanding of MLKL signaling. We also now add to these findings by showing that MLKL may regulate fatty acid accumulation in hepatic cells leading to defective expression of key lipid metabolism genes, supporting our *in vivo* data

We greatly appreciate the fact the reviewers pointed out the robust nature of our study, the importance of our direct comparison of RIPK3 and MLKL deficient mice and the novelty of our findings on noncanonical MLKL-mediated effects on lipid metabolism. As such, we believe our manuscript will be of great interest to cell death, inflammation and metabolism researchers.

Reviewer #1

R#1. *The overall message of this article is intriguing and clearly highlighting an interesting scientific novelty which includes MLKL role in obesity independently of RIPK3 seemingly via transcriptional upregulation or/ and potentially via regulating metabolic organelles and lipid receptors. While I do think that this, needs further future investigations, I also believe that both the cell death field and the metabolism field would benefit from this piece of evidence being published in this journal. The manuscript takes almost a Pindaric inspiration, starting with the RIPK3-caspase-8 role in signaling independent of its function to activate NLRP3, then going through the role of Casp8 in obesity which has been largely published and then ending with the role of MLKL in obesity and which is the novel part of the story. While I think that, it is nice that the authors gave such a good set up to their study, I do find that the first half of the manuscript is something that was quite known already. It is true that there are many clarifications in their approach however most of this could be included in supplementary material. This is because the real interest comes with MLKL and in the first part the authors never include MLKL into their story or systems, cellular or animal. The MLKL dependent role in obesity is very intriguing and opens many questions which are highlighted by the authors.*

Response 1.

We greatly appreciate the reviewer's comments on the thoroughness of our work that documents the role of RIPK3-caspase-8 and MLKL signaling in obesity and NAFLD. We acknowledge that the role of caspase-8 has been addressed in other studies but believe our study adds significantly by pinpointing that RIPK3-caspase-8 signaling contributes to the transcriptional regulation of NLRP3 inflammasome components/cytokines, direct IL-1 β cleavage and death in macrophages in response to LPS and palmitate. However, as acknowledged by the reviewer the major impact of this work stems from our findings on the effect of MLKL on obesity and non-alcoholic fatty liver disease (NAFLD), so we have moved much of the *in vitro* LPS/palmitate macrophage data

(**Revised Fig S1 [NLRP3] and S2 [RIPK3, MLKL, RIPK3/casp8]**) and *in vivo* HFD *casp8^{LysMcre}Ripk3^{-/-}* data (**Revised Fig S3-S5**) to the supplementary data.

It is important to note that our study starts off by asking whether dietary fatty acids trigger extrinsic apoptotic or necroptotic signaling to induce NLRP3 inflammasome activation in macrophages (Lawlor et al *Nat Commun* 2015 PMID: 25693118; Conos et al *PNAS* 2017 PMID: 28096356), which has been intensely studied in several infectious and non-infectious diseases, as referenced (Lines 102-107) in the revised manuscript. Importantly, we genetically show that neither caspase-8 nor MLKL are obligatory for NLRP3 inflammasome activity and that RIPK-caspase-8 signaling is required for effective inflammasome priming (*i.e.*, NLRP3 and pro-IL-1 β induction) in response to TLR2/4 ligation and palmitate exposure (**Revised Fig. 1A & Fig S2E**). We have now amended the results section and provide a better rationale for examining NLRP3 inflammasome activity to palmitate and the role of core apoptotic and necroptotic machinery (Lines 119-125).

R#1. *The authors should at least attempt to utilize a cellular system to support some of their speculation with regards to the way that MLKL might mediate this transcriptional regulation. Does it translocate to the nucleus? Is it associated to a transcription factor?*

Response 2.

In our discussion, we speculated that MLKL may impact lipid trafficking/accumulation and perturb fatty acid and triglyceride responses downstream of key transcriptional pathways in tissue cells, independent of canonical RIPK3 activity (**Revised Fig 5 and 6 and Fig S9 & NEW DATA Fig S10, reproduced below for the reviewer's convenience**). To provide further mechanistic insights into how MLKL regulates responses to saturated fatty acids, we generated 2 MLKL-deficient Hepa1-6 cells lines using CRISPR/Cas9 gene editing (**NEW DATA Fig S12A, B; reproduced below**). As these cells lack RIPK3 expression, mimicking primary mouse hepatocytes that display epigenetic silencing of RIPK3 (Preston et al *Gastroenterology* 2022 PMID: 36037995), we also reconstituted cells with a stably integrated, doxycycline (DOX)-inducible, FLAG wildtype RIPK3 to facilitate canonical RIPK3 signaling (Frank et al *iScience* 2022 PMID: 35800780). As expected, WT and *Mkl1^{-/-}* Hepa1-6 cells were sensitive to RIPK1-dependent apoptosis (TNF + Smac mimetic; TS) and resistant to RIPK3-MLKL necroptotic cell death (TNF/Smac mimetic/Q-VD-OPh; TSQ). Importantly, induction of RIPK3 expression with DOX treatment restored necroptosis in WT but not MLKL-deficient cells (**NEW DATA Fig S12A, B**).

Using this system, we next assessed whether palmitate may restore RIPK3 expression to drive canonical MLKL-mediated hepatic cell death. Remarkably, we found palmitate failed to induce RIPK3 expression, and we observed comparable cell death at 16 hours in WT and *Mkl1^{-/-}* Hepa1-6 cells to increasing concentrations of BSA-PA, in the presence or absence of RIPK3 (**NEW DATA Fig S12 C, D**). Importantly, while MLKL deficiency did not impact cell death, we did observe reduced lipid accumulation in *Mkl1^{-/-}* Hepa1-6 cells upon stimulation with lower, less lipotoxic doses of BSA-PA, which was also independent of RIPK3 expression (**NEW DATA Fig S12E**). We next analysed if reduced lipid uptake in MLKL-deficient hepatic cells was associated with altered gene transcription basally and/or in response to palmitate and found reductions in mRNA levels for genes involved in lipid regulatory processes – transcription, lipid uptake, biogenesis, triglyceride synthesis, peroxisomes and lipid droplet formation (**NEW DATA Fig S12F-K**). Collectively, these results, combined with our *in vivo* HFD data, suggest that MLKL may regulate lipid accumulation/trafficking resulting in changes in the expression of lipid regulatory genes, largely in the absence of RIPK3.

As the reviewer asked whether, based on its propensity to bind intracellular organelle membranes, if MLKL could be targeting the nucleus to regulate a specific transcription factor, we performed fractionations to isolate nuclear and cytoplasmic proteins in palmitate-treated Hepa1-6 cells. As a positive control we included necroptotic stimuli (TSQ) to induce translocation of RIPK1, RIPK3 and MLKL to the nucleus, where reports suggest that either active, phosphorylated proteins translocate to the nucleus prior to necrosome assembly (Yoon et al *CDD* 2016 PMID: 26184911) or RIPK3 and MLKL are activated in the nucleus to form an active complex (Weber et al *Commun Biol* 2018 PMID: 30271893). Intriguingly, while we did detect MLKL in the nuclear extract in response to palmitate, as well as to TSQ (**see below Rebuttal Figure 1**), these experiments were made challenging by the large amount of contaminating DNA in hepatocytes and were inconclusive. Furthermore, in parallel experiments, we determined that RIPK3 is not expressed in these hepatic cells (**Fig S12A, C**), nor could we detect RIPK3 or phosphorylated MLKL in hepatocytes in our HFD-fed mouse livers (**See Response 3 and Fig S11A below**). While it is still possible that MLKL translocates to the nucleus via a

noncanonical signal to regulate transcription, akin to a recent report that MLKL binds RNA-binding protein RBM6 to stabilize adhesion molecule mRNA levels (Dai et al *CDDis* 2020 PMID: 32332696), we believe defining this mechanism is beyond the scope of this already comprehensive study. In the future, we plan to investigate whether and how MLKL co-localizes to organelles, in the absence of RIPK3, using our cell lines and primary tissue cells, and whether this impacts lipid biology.

Figure S10

Figure S10 (NEW DATA) Expression of fatty acid and cholesterol lipid metabolism regulatory genes in ageing ND and HFD-fed Mlkl^{-/-} livers.

(A, B) qRT-PCR measurement of relative levels of (A) cholesterol metabolism regulatory genes *Srebp2*, *Hmgcr*, *Hmgcs2* and *Ldlr* and (B) fatty acid synthesis transcription factor *Srebp1* in ND- and HFD-fed WT and Mlkl^{-/-} liver tissue after 23-25 weeks of diet (fold change over WT ND). Data show the mean \pm SEM, $n \geq 6$ mice per group pooled from 3 experiments. One-way ANOVA followed by Tukey's multiple comparison test, ** $P < 0.01$, *** $P < 0.001$. (C) Liver lysates from WT and Mlkl^{-/-} mice fed a ND or HFD for ~25 weeks were analyzed by immunoblot for FABP4. $n = 3$ mice per group, each lane represents an individual mouse. FABP4 levels were analyzed by densitometry and normalized to Actin and expressed as a fold change over WT ND liver lysates. Results are presented as the mean \pm SEM. One-way ANOVA followed by Tukey's multiple comparison test, ** $P < 0.01$.

Figure S12

Figure S12. MLKL regulates saturated fatty acid accumulation and transcriptional responses in hepatic cell lines in a RIPK3-independent manner.

(A, B) WT and *Mlkl*^{-/-} Hepa1-6 cells (Clone #1 and #3) were treated with Doxycycline (DOX, 100 ng/ml) for 3 h, as indicated, to induce RIPK3 expression and then treated with TNF (T, 100 ng/ml), Smac-mimetic 711 (S, 1 μM) and pan-apoptotic caspase inhibitor Q-VD-OPh (Q, 40 μM) for 12-16 h. (A) Immunoblots were performed on cell lysates for relevant proteins. The results are representative of 2 independent experiments. < Reprobe of MLKL membrane, * Non-specific bands. (B) Cell death was measured via flow cytometric analysis of PI uptake. Data show the mean ± SD, n = 3 replicates from 1 of 3 experiments. One-way ANOVA followed by Tukey's multiple comparison test, ****P <

0.0001. (C-E) WT and *Mkl1*^{-/-} Hepa1-6 cells were treated with DOX (100 ng/ml) for 3 h to induce RIPK3, as indicated, and media replaced. Cells were then stimulated with 100-300 μ M palmitate conjugated to BSA (BSA-PA) or BSA (equivalent to highest BSA-PA) for 16-24 h. (C) Cell lysates were subjected to immunoblot for relevant proteins. Representative of 2 independent experiments. < Reprobe of MLKL membrane. (D) Cell death was measured via flow cytometric analysis of PI uptake. Data show the mean \pm SD, n = 3 replicates from 1 of 3 experiments. One-way ANOVA followed by Tukey's multiple comparison test, **P < 0.01, ****P < 0.0001. (E) Lipid accumulation was measured by BODIPY staining and flow cytometric analysis. Data are presented as the mean \pm SD, n = 4 technical replicates pooled from 2 independent experiments. One-way ANOVA followed by Tukey's multiple comparison test, *P < 0.05, **P < 0.01, ***P < 0.001, ****P < 0.0001. (F-L) WT and *Mkl1*^{-/-} Hepa1-6 cells were treated with BSA-PA (200 μ M) or BSA equivalent for 16 h and relative levels of lipid metabolism related genes measured by qRT-PCR. Data shown are the mean \pm SD, n = 4 replicates representative of 1 of 2 independent experiments. One-way ANOVA followed by Tukey's multiple comparison test, *P < 0.05, **P < 0.01, ***P < 0.001, ****P < 0.0001.

Rebuttal Fig 1. Translocation of MLKL to the nucleus. Hepa1-6 cells were treated with 200 μ M PA-BSA or BSA equivalent, or with necroptotic stimuli TSO (TNF 100 ng/ml + Smac-mimetic 711 1 μ M + Q-VD-OPh 40 μ M) for 4 h. Nuclear and cytoplasmic extracts were prepared from whole cell lysates (WCL) and analysed by immunoblot for RIPK1, MLKL, GAPDH (cytoplasmic marker) and H3Cit (nuclear marker). 1 of 2 repeats.

R#1. The authors also comment on the fact that MLKL is found bound to lipids and cite, among many, Quarato et al. from the Green lab. However, it is important to specify that such ability of MLKL to bind the lipids in the membrane or lipids in general, necessitates the initial RIPK3 phosphorylation which allows the conformational change needed for the 4HBD to form oligomeric structures. It is really unclear to me how the authors envisage the ability of MLKL to bind lipids independently of RIPK3 or independently of the only known phosphorylation event capable of driving its open conformation. On this point it would be nice if the authors could determine if MLKL localizes in liver adipocytes or around them or from extracts? What about RIPK3?

Response 3. Yes, in general the kinase activity of RIPK3 is needed for MLKL binding to phosphatidyl-inositol phosphates (PIPs) in membranes. However, as we highlight in the discussion, there are reported cases of noncanonical MLKL activity, including RIPK1-mediated activation of necroptotic MLKL signaling (Gunther et al *JCI* 2016 PMID: 27756058). MLKL is also reported to undergo noncanonical phosphorylation at serine 441 (by an unknown kinase) to allow binding of sulfatide and myelin sheath breakdown in Schwann cells, independent of RIPK3 and RIPK1 (Ying et al *Mol Cell* 2018 PMID: 30344099). Furthermore, MLKL can bind early endosomes to promote endocytic trafficking and vesicle release (Yoon et al *Immunity* 2017 PMID: 28666573; Rasheed et al *Arterioscler Thromb Vac Biol* 2020 PMID: 32212851) and target the autophagolysosome to inhibit autophagic flux (Wu et al *J Hepatol* 2020 PMID: 32220583; see **Response 4**) in a RIPK3-independent manner

To try and address this critique, we firstly performed western blots to determine RIPK3 and MLKL expression

in the liver in response to chronic HFD feeding. While we could detect MLKL in WT liver tissues, we failed to detect significant levels of RIPK3 or phosphorylation of MLKL at serine 345 in HFD-fed mice (**NEW DATA Fig S11A**), which is in line with increasing reports epigenetic silencing of RIPK3. To enhance our detection of RIPK3 at a cellular level, we next adopted our recently published protocol for RIPK3 staining of paraffin embedded liver sections (Chiou et al *EMBO Mol Med* 2024 PMID: 38750308). Noting that we could not analyse MLKL expression in our paraffin embedded liver sections as we have previously found that 3 anti-mouse MLKL antibodies are highly non-specific in most tissues, including the liver, by immunohistochemistry (Chiou et al *EMBO Mol Med* 2024). In addition, our tissue processing protocol precluded our use of an optimized pMLKL Ser345 immunostaining method that was recently published by the Liccardi laboratory (Kelepouras et al *CDD* 2024). However, matching our *in vitro* data (see **Response 2**), we failed to detect RIPK3 in hepatocytes in liver sections from WT or *Mlkl*^{-/-} mice fed a ND or HFD. Instead, we observed RIPK3 expression is largely restricted to liver macrophages and crown-like structures that form around lipid droplets upon HFD-feeding, as well as in VAT macrophages, which we believe (based on our data, **Figs 1, 2 and S5F-H**) signal via caspase-8 to drive inflammation (**NEW DATA Figure S5A, S6I; and S11B and Rebuttal Fig 3 below**).

Figure S11

NEW DATA Fig S11. RIPK3 expression is not induced in liver tissue cells upon ageing or HFD feeding. A. Liver lysates from WT and *Mlkl*^{-/-} mice fed a ND and HFD for ~25 weeks were subjected to immunoblot for the indicated proteins. *n* = 3 mice per genotype and diet. **B.** Representative RIPK3 immunostaining in liver and VAT sections from WT and *Mlkl*^{-/-} ND and HFD fed mice. *n* = 9-10 mice per group examined. Scale bar 100 μ m. Box 200x magnification (bottom row).

Rebuttal Fig 3. Further examples of RIPK3 staining in WT and MLKL-deficient ND and HFD livers. Representative RIPK3 immunostaining in liver sections from WT and *Mlkl*^{-/-} ND and HFD fed mice. Scale bar 100 μ m. Box 200x magnification.

R#1. Additionally, MLKL has been shown to be ubiquitylated on different sites leading also to different outcomes: degradation (Silke lab), cell death (Meieir Lab) and autophagy (Wallack lab). Considering the autophagy angle and also the role in exosome release (also from the Wallack lab) I wonder if the authors could provide any evidence of MLKL ubiquitilation and/or autophagy activation that would justify the defective uptake of fatty acids. This might still remain correlative, however would provide support to some of the conclusions that have been discussed

Response 4. We agree that investigation of these pathways would be interesting, particularly autophagy as the Nagy group reported a RIPK3-independent role for MLKL in inhibiting autophagic flux to palmitic acid *in vitro* and upon a high fat, fructose and cholesterol diet *in vivo* (Wu et al *J Hepatol* 2020). However, as we found no gross increases in endoplasmic reticulum (ER) stress (e.g. CHOP, Bip, ATF4) or autophagy (ATG5, ATG7) related gene expression in livers from WT and *Mlkl*^{-/-} mice fed a ND or HFD by RNAseq, we chose to focus our study on deciphering how MLKL perturbs lipid metabolism. We have now performed western blots on liver tissue lysates from our starved (>4 h) ND- and HFD-fed mice to see if MLKL inhibits autophagic flux and induces ER stress in our model. In line with Wu et al, we saw signs of impaired autophagic flux (i.e. LC3 II/I ratio and p62) in WT HFD-fed livers but not in HFD-fed *Mlkl*^{-/-} livers, but impacts to ER stress (p-eIF2 α /eIF2 α ratio) were less clear (**Rebuttal Fig 2**). Unexpectedly, we also observed elevated LC3 II and p62 levels in our ND-fed *Mlkl*^{-/-} liver that contradicts MLKL's suggested role in blocking autophagy. Consequently, while promising, our results are inconclusive and therefore warrant further detailed investigation, beyond the scope of this study.

Rebuttal Fig. 2. Liver lysates from WT and *Mlkl*^{-/-} mice fed a ND or HFD for ~25 weeks were analysed for levels of autophagy regulators (p62, LC3B I conversion to LC3B II) and ER stress response (peIF2 α /eIF2 α ratio). n = 3 mice per group. Representative of 1 of 2 experiments. * non-specific band. Densitometry was performed and protein levels normalised to GAPDH and then expressed as a fold change over WT ND (mean \pm SEM).

R#1. I am not convinced that the data provided can completely exclude the dependency from RIPK3. This is

because neither the MLKL/RIPK3 DKO nor the RIPK3 KO alone has been employed to show the differences between these two animal models at least in terms of transcription. Additionally, it would be interesting to compare the *Casp8^{LysMcre}/RIPK3^{-/-}* with the *Casp8^{LysMcre} / Mlkl^{-/-}*. According to the authors conclusions the distinct roles of *Casp8* and MLKL should then provide a further advantage? I completely understand that this might be a rather long experiment that would involve several animal crosses, hence perhaps the utilization of caspase inhibitor in the background of MLKL deficiency could be an option.

Response 5. While we agree that examination of *Casp8^{LysMcre} Mlkl^{-/-}* and MLKL/RIPK3 double knockout mice would be interesting, these experiments are well beyond the scope of this already detailed study, as it would take up to 2 years to generate (or rederive knockout lines) and the model itself take 33 weeks from birth (+ readouts). Likewise, use of a caspase inhibitor would require considerable optimization and may be confounded by differences in cell-specific targeting and effects. For example, a recent report suggested that the caspase-8 inhibitor z-IETD-fmk can induce RIPK3-dependent, MLKL-independent cytokine production from neutrophils *in vivo* during pathogen signaling (PMID: 37390829). It is important to note, our new data presented in **Response 2** and **3** suggest MLKL may signal in the absence of RIPK3 in liver cells.

Reviewer #2

R#2. *In the manuscript by Tye et al., the authors investigated the roles of key cell death players in high fat diet-induced inflammation and lipid metabolism using genetically modified mice. The manuscript appears to be composed of two distinct sections. In the first part, the authors reported that the absence of both Casp8 and Ripk3 conferred resistance to LPS/palmitic acid-induced inflammation in vitro and high fat diet-induced steatosis in vivo. In the second part, the authors showed that the absence of Mlkl also reduced lipid accumulation through the transcriptional regulation of lipid metabolism. Although experiments are all conducted in a scientifically sound manner, several critical data are missing. Moreover, the phenotypes of Mlkl-deficient mice induced by high fat diet was previously published (Saeed et al., PMID: 31132314, DOI: 10.1111/jgh.14740), and more in-depth analysis of molecular mechanisms may be necessary. Therefore, I would suggest that following points should be addressed to improve the scientific quality of this manuscript.*

Response 6.

We thank the reviewer for acknowledging our experiments were conducted in a scientifically sound manner. Our manuscript shows that MLKL also impacts adiposity in ageing and HFD-fed mice. Moreover, we show that MLKL is not as important for regulating HFD-induced tissue inflammation, compared with RIPK3-caspase-8 signaling. MLKL, instead, appears to regulate lipid handling and metabolism in liver cells, independent of RIPK3 (see **Response 2**)

R#2. *Although Casp8, Ripk3, and Mlkl are all critical proteins of cell death machinery, the evaluation of cell death is only presented for BMDMs in vitro (PI uptake), but not for genetically modified mice in vivo. I think that Figs 3 and 5 should also contain the photographs and quantitative data of cell death. TUNEL staining may detect various modes of cell death (apoptosis, necroptosis, or pyroptosis). Staining of cleaved caspase-3 may be useful to evaluate whether apoptosis still occurs in the absence of Casp8 and Ripk3.*

Response 7.

Unfortunately, the paraffin blocks for our *Casp8^{LysMcre} Ripk3^{-/-}* and *Casp8^{lox/lox} Ripk3^{-/-}* HFD experiments were discarded by our histology facility during the COVID-19 pandemic thus preventing us from performing TUNEL (dead cells) or M30 (apoptotic hepatocyte) staining. We did attempt to assess cleaved caspase-3 staining on existing ND and HFD VAT and liver sections. Intriguingly, we observed increased cleaved caspase-3 staining associated with crown-like macrophage structures in WT HFD-fed VAT compared to *Casp8^{LysMcre} Ripk3^{-/-}* and *Casp8^{lox/lox} Ripk3^{-/-}* (NEW DATA Fig S5A and S6I). However, our attempts to quantify cleaved caspase-3 in liver sections, using our published imaging analysis (Chiou et al *EMBO Mol Med* 2024), revealed no overall differences in cleaved caspase-3 positivity in *Casp8^{LysMcre} Ripk3^{-/-}*, *Ripk3^{-/-}* or *Mlkl^{-/-}* HFD fed mice, compared to WT (**Rebuttal Fig 4**), although, we believe that quantification is likely impacted by the severe steatosis and degenerative ballooning seen in WT livers.

Rebuttal Fig 4. Cleaved caspase-3 staining in ND and HFD WT and MLKL-deficient livers

Representative microscopy images of HFD-fed control and mutant liver sections stained for levels of F4/80 (macrophage marker) and cleaved caspase-3 (apoptotic cells) and quantification of the cleaved caspase-3+ positive cells/area (%) per tissue area quantified.

R#2. Also, as liver fibrosis was evaluated by Sirius red staining, representative images of Sirius red photographs may be necessary for Figs 3 and 5.

Response 8.

Please find examples of Sirius red staining (NEW DATA, reproduced above) now included in the revised manuscript as Fig S5D and Fig S6I to complement Fig 2J, K and Fig 4E, F, respectively.

R#2. In Figures 1F and S1J, it is formally necessary to include the Western blots of MLKL and RIPK3 to indicate the absence of these proteins in BMDMs. If possible, lack of these proteins should be confirmed in the Western blots of fat and liver tissues.

Response 9.

The MLKL-, RIPK3- and RIPK3/Casp8- deficient mice used in this study have been validated to be knockouts in several studies, including in BMDMs isolated from these animals (Allam et al *EMBO Rep* 2014 PMID: 24990442; Lawlor et al. *Nat Commun* 2015, Vince et al *Cell Rep* 2018 PMID: 30485804). We now include a western blot confirming that our knockout mice lack relevant proteins (NEW DATA Fig S2A; reproduced below).

A

NEW DATA Figure S2A. WT, Mlkl^{-/-}, Ripk3^{-/-} and Ripk3^{-/-}Casp8^{-/-} BMDMs were primed with LPS for 3 hours, as indicated, and treated with BSA and BSA-PA (600 μM) for 18 hours. Cell lysates were analysed for relevant proteins. n = 1 experiment

R#2. In the GO analysis of Figure 6C, the authors focused on the downregulation of fatty acid metabolism in Mlkl KO mice fed normal diet. However, it appears equally important that cholesterol biosynthesis is upregulated in Mlkl KO mice under normal diet. These results suggest that both fatty acid and cholesterol metabolic pathways are altered in the absence of MLKL. Although the amounts of serum and liver cholesterol contents shown in Fig 7 may not reflect the transcriptional upregulation of cholesterol biosynthesis, it may also be insightful to examine the expression of key genes in cholesterol metabolism, such as SREBP2, HMG-CoA synthase, or HMG-CoA reductase.

Response 10.

We thank the reviewer for making this point and have now examined cholesterol metabolism related genes, as requested, and included this data in our revised manuscript (**NEW DATA Figure S10A, see Response 2**). Interestingly, we saw no significant difference in *Srebp2*, *Hmgcr*, *Hmgcs2* and *Ldlr* in Mlkl^{-/-} fed a ND or HFD. (NB. *Hmgcs2* and *Ldlr* did trend down in HFD-fed KO mice).

R#2. It has been reported that extracellular addition of palmitic acid induces ER stress. Please discuss the potential contribution of ER stress in this experimental system.

Response 11.

We have now amended the text to introduce the proposed mechanisms surrounding how saturated fatty acids induce may drive hepatocyte cell death in NAFLD, including autophagy and ER stress (Lines 482-485). Beyond this, we have highlighted in the discussion the potential that, independent of RIPK3, MLKL may inhibit autophagy to drive ER stress and lipotoxicity (Lines 485-488).

R#2. Minor points:

Response 12.

- In the titles of GO analysis graphs in Figs 6 and S4, "FC" should be "logFC". This has been corrected.

- In Line 117, reference of Fig S1J should be moved to Line 146.

This has now been referred to correctly in the revised manuscript as (**Fig S2D-F**) on line 159.

- In Lines 145-6, the authors described that "appreciable IL-1β(p17) secretion in Ripk3^{-/-}Casp8^{-/-}BMDMs". However, IL-1β in supernatant is very low in BMDMs derived from Ripk3^{-/-}Casp8^{-/-} mice compared to those from mice of the indicated genotypes (Fig S1J). Is this interpretation correct?

This was to indicate that IL-1β was being activated still in the absence of RIPK3 and caspase-8. We have amended to state that "low levels of mature IL-1β p17 secretion" were detectable (Line 158).

- In Line 146, reference may be "Fig. S1I to J".

We have corrected this reference in the revised manuscript to Fig S2D-F on line 159.

- In Line 149, reference may be "Fig 1F and H and S1K and L".

We have corrected this in Line 162 of the revised manuscript to **Fig 1D and S2G**.

- The reference of Fig S2O (gating strategy of liver FACS) is missing. They may be in Line 207 and Line 246. FigS2N (gating strategy of VAT FACS) may also be referenced in Line 240.

Apologies we have now referenced the gating strategy (now **Fig S5C VAT and S5E Liver**) in lines 229 and 235 of the revised manuscript.

- In Line 284, the reference may be "Fig S4C and E and Fig S5a" for HFD-fed mice.

This has now been corrected in the revised manuscript to (**Fig S8D and E and S9A**) in line 310.

- In Line 768, "J.H.M" may be "J.M.H" (Joanne M Hildebrand?).

This has now been corrected.

Reviewer #3:

R#3. Authors provide evidence of the role of Casp8-Ripk3 axes in macrophages in the development of NAFLD in the context of obesity and a non canonical role of MLKL independent of its classical cell death role. The study is relevant to the field and its publication increases the evidence of the role of cell death components in metabolism. Authors uncovered that RIPK3-caspase-8 signaling is not mandatory for NLRP3 inflammasome activation in LPS treated macrophages exposed to palmitate, but instead caspase-8 regulates inflammatory gene transcription, direct IL-1b proteolysis, and to a lesser extent cell death in LPS and palmitate treated macrophages. In addition, their in-vivo flow cytometry data reveal less recruitment of inflammatory immune populations to the liver of obese mice, improvement of several metabolic parameters and reduce steatosis. On the other hand, the manuscript shows a new role of MLKL in lipid metabolism that seems to be independent of its cell death functions. Data shows that MLKL deletion prevents liver steatosis, reduces adipocyte size and gain weight, while at the same time reduces systemic inflammation independently of immune cell activation of MLKL. Revealing the vital role for MLKL in regulating lipid uptake, transport and metabolism. Authors also mention the need for future studies investigating the proximity of MLKL to specialized metabolic organelles and examination of the expression and function of lipid receptors/transporters.

R#3. Throughout the first figures it is evident that the protection from obesity-induced metabolic dysfunction is afforded by RIPK3 deficiency rather than by the deficiency of Casp8 in myeloid cells. Therefore, the conclusion of their data in Figures 2 and 3 needs dramatic revision (including the title). If the authors cannot look into metabolic features in Casp8 in myeloid KO mice because of the leaky model, then it is very difficult to draw any conclusion on the Casp8 angle given the data provided. This also means that the findings here are in line with the previous findings by Gautheron et al. Since the evidence on the role of RIPK3 and MLKL in obesity-induced metabolic dysfunction is indeed conflicting, this side-by-side comparison between RIPK3 and MLKL deficiencies is valuable, despite the fact that there is no novelty on the phenotype of RIPK3 knockout mice upon HFD.

Response 13. We firstly thank the reviewer for pointing out that our direct comparison of RIPK3 and MLKL KO lines is valuable to the research community. We agree that protection is largely afforded by RIPK3 deficiency alone and acknowledge that the kinase activity of RIPK3 itself can drive transcription (PMID: 27396959, 25367573). Nevertheless, based on the strong link between active caspase-8 and TLR-MyD88-mediated transcription (PMID: 32971525), we still believe our data in the *Casp8^{LysMcre}Ripk3^{-/-}* mouse suggests that *in vivo* a RIPK3 platform likely regulates caspase-8 activity in myeloid cells to drive death and inflammation-related changes to a HFD, which is in line with our previous work in endotoxin and arthritis models (Allam et al *EMBO Rep* 2014; Lawlor et al *Nat Commun* 2015). We have reframed our findings to acknowledge the protection conferred by RIPK3, including retitling the manuscript, yet we maintain that additive effects are observed in *Casp8^{LysMcre}Ripk3^{-/-}*, which are directly attributable to the robust loss of caspase-8.

Regarding the paper by Gautheron et al, they found that global RIPK3 deficiency was associated with increased obesity, insulin resistance and VAT inflammation to a choline-deficient diet (Gautheron et al 2016), while we report reduced inflammation in RIPK3-deficient mice that is further enhanced with caspase-8 loss. Furthermore, our new data suggests that RIPK3 is not detectable in hepatocytes but rather in macrophages and infiltrating cells in the liver and VAT, so differences are likely to be related to macrophage biology (**NEW**

DATA Figure S11B).

R#3. *Another concern is the visible lack of connection between Figure 1 and the rest of the paper. If the authors see a clear connection this should be better conveyed in the text and perhaps some experimental evidence of this connection should be provided.*

Response 14. We have now simplified and revised the paper to better introduce the rationale for Figure 1 and attempted to connect these findings better with our analysis of inflammasome-dependent IL-1 β secretion from VAT explants (Revised **Fig 2I and 4D**) and gene expression in the liver (**Fig 4H** and **NEW DATA Fig S5H**).

R#3. *Specific comments, Please clarify what is meant here:*

157 and caspase-8 deletion is inefficient

158 upon myeloid-specific deletion (Lysozyme M Cre transgene) (Kang et al., 2004), we chose to
159 compare mice lacking caspase-8 conditionally in myeloid cells on a RIPK3-deficient background
160 (Casp8LysMcreRipk3^{-/-}) (Vince et al., 2018) that will have a more robust loss of RIPK3-caspase-8
161 signaling in myeloid cells compared to RIPK3 deficiency alone.

Are you comparing with Casp8Lysmye alone? Or with RIPK3 KO? If the latter, then this phrase is superficial since in this case Casp8 is wt...

Response 15. Apologies for the confusion. We have now clarified our meaning that genetic deletion of caspase-8 in myeloid cells is sub-optimal due to the necroptotic loss of myeloid progenitors in deleted cells and escape of non-deleted cells. However, deletion of RIPK3 prevents this progenitor loss and allows for more efficient deletion of caspase-8. This affords us the opportunity to compare that *Ripk3^{-/-}* mice can have blunted caspase-8 signaling in myeloid cells, as discussed in **Response 14**.

R#3. *Fig 1 legend: (C) what are the technical replicates for the authors? Are each of the dots BMDMs isolated from 4 or 5 different mice or just the same cells treated in different wells? If they are not individual mice then the representation is wrong, technical replicates need to be expressed as average, and the average is the result of one individual experiment. On the contrary, each mouse can be considered as an individual experiment, so if this are cells isolated from 4 different mice (and kept separated), then these are individual experiments.*

Response 16. We thank the reviewer for pointing this out and had thought viewing the technical variability within experimental replicates would be valuable. However, we have now presented all relevant data sets as an average of technical replicates from individual experiments and also generated a new data set of our key data set using 3 mice to show the robustness and reproducibility of our findings (**NEW DATA Fig 1B-D and Fig S2D, F and G**)

Figure 1 (revised with NEW DATA). Caspase-8 activity contributes to LPS and palmitate-induced NLRP3 inflammasome priming, IL-1β activation and macrophage cell death. (A-D) WT, *Mlkl*^{-/-}, *Ripk3*^{-/-} and *Ripk3*^{-/-}*Casp8*^{-/-} BMDMs were primed with or without LPS (50 ng/ml) for 3 h and treated with 600 μM PA-BSA or BSA alone, as indicated, for 18-20 h. (A) Cell lysates and supernatants were analyzed by immunoblot for specified proteins. The results shown are representative of 2 independent biological experiments. (B) IL-1β and (C) TNF levels were measured in cell supernatants by ELISA (NEW DATA). Data shown are the mean ± SEM of n = 3 biological replicates and are representative of at least 4 independent experiments. One-way ANOVA followed by Tukey's multiple comparison test, ****P < 0.0001. (D) Cell viability was assessed by PI uptake and flow cytometric analysis (NEW DATA). Data shown are the mean ± SEM of n = 3 biological replicates and are representative of at least 4 independent experiments. one-way ANOVA followed by Tukey's multiple comparison test, ****P < 0.0001.

Figure S2

Figure S2. Defective caspase-8-mediated inflammasome priming reduces palmitate-induced NLRP3 inflammasome activation.

(A) WT, *Mlkt*^{-/-}, *Ripk3*^{-/-} and *Ripk3*^{-/-}*Casp8*^{-/-} BMDMs were primed with or without with LPS (50 ng/ml) for 3 h and treated with 600 μM PA-BSA or BSA alone, as indicated, for 18-20 h. Cell lysates were subjected to immunoblot for RIPK3, caspase 8 and MLKL. Results shown are representative of 1 experiment (NEW DATA). (B, C) WT, *Mlkt*^{-/-}, *Ripk3*^{-/-} and *Ripk3*^{-/-}*Casp8*^{-/-} BMDMs were primed with LPS (50 ng/ml) for 3 h and treated with nigericin (10 μM) for ~ 45 min. (B) IL-1β levels were measured in cell supernatants by ELISA. (C) Cell

viability was assessed by PI incorporation and flow cytometric analysis. Data show the mean \pm SD of $n = 3$ independent biological experiments showing the average of two replicates per experiment. One-way ANOVA followed by Tukey's multiple comparison test, $**P < 0.01$. **(D-G)** WT, $Mlkl^{-/-}$, $Ripk3^{-/-}$ and $Ripk3^{-/-}Casp8^{-/-}$ BMDMs were primed with or without Pam3Cys (500 ng/ml) for 3 h and treated with 600 μ M PA-BSA or BSA alone, as indicated, for 18-20 h. **(D)** IL-1 β and **(F)** TNF levels were measured in cell supernatants by ELISA (**NEW DATA**). Data are the mean \pm SEM of $n = 3$ biological replicates and are representative of at least 4 independent experiments. One-way ANOVA followed by Tukey's multiple comparison test, $*P < 0.05$, $***P < 0.001$, $****P < 0.0001$. **(E)** Cell lysates and supernatants were analyzed by immunoblot for specified proteins. The results shown are representative of 2 independent biological experiments. **(G)** Cell viability was assessed by PI uptake and flow cytometric analysis (**NEW DATA**). Data show the mean \pm SEM of $n = 3$ biological replicates and are representative of a least 3 independent experiments. One-way ANOVA followed by Tukey's multiple comparison test, $****P < 0.0001$.

R#3. Fig 1 E: ANOVA analysis. Please explain the question that you wanted to answer when you compared the MLK KO against RIPK3/casp8 KO in the Palmitic acid treatment. Which questions did you have behind the design of this test? If this is a 2-way ANOVA, which are the 2 parameters that you are comparing? Genotype among the same treatment is 1-way (and you can select the comparisons pairs according to your experimental design), Same genotype, two different treatments, is also one comparison. For (G) the statistical test seems to be different.

Response 17. Apologies, this was an oversight and we have now performed a one-way ANOVA analysis for relevant comparisons. The rationale for comparing between WT, RIPK3, MLKL and RIPK3/casp8KO in the palmitate treatment was to discern which genotype was defective in their response and ascertain whether MLKL or caspase-8 regulated IL-1 β responses

R#3. Fig 2 D. the authors mention that RIPK3 KO mice tend to be more obese (line 172). This is not shown in any of the figures in the paper, including supplementary. The meaning of this conclusion should be better explained or revisited.

Response 18. Agreed, there was only a trend based on % organ weight normalised to body mass (revised **Fig S3G**). We have removed reference to this in the text.

R#3. Please clarify the following

176 No overall difference in weight gain or organ mass was observed in genetic control mice
177 lacking caspase-8 only in myeloid cells ($Casp8^{LysMcre}$) that display inefficient deletion (fig. S2G 178 and H) (Kang et al., 2004; Lawlor et al., 2015), suggesting that removal of upstream RIP kinase 179 machinery is required to see the full impact of apoptotic caspase-8 activity.
Do you mean that upstream RIP kinase machinery is required to see the full impact of the loss of caspase 8 activity? So, the prevention of apoptosis? In any case, as mentioned in the general statement, it seems that RIPK3 deficiency is playing a major role in your assessment.

Response 19. As discussed above in **Response 15**, caspase-8 deletion is sub-optimal in myeloid cells under a LysM cre transgene, which is improved with the concurrent deletion of RIPK3. We acknowledge that RIPK3 is playing an important role but our data shows that there are additive effects with loss of caspase-8. The phenotype observed in the $Casp8^{LysMcre}$ mice is largely reflective of the model's poor deletion of caspase-8, rather than being attributable to actual loss of caspase-8, which reinforces our comparisons of $Casp8^{LysMcre}Ripk3^{-/-}$ and $Casp8^{lox/lox}Ripk3^{-/-}$ mice.

Comment: explain why $Casp8^{LysMcre}Ripk3^{-/-}$ mice can be better in glucose clearance but no differences are shown in insulin tolerance or leave it as an open question. This observation might be extremely important to understand the phenotype, since in general rules, mice fed with HFD become first insulin resistant and latter, glucose intolerant.

Response 20.

While $Casp8^{LysMcre}Ripk3^{-/-}$ mice fed a HFD did perform better than $Casp8^{lox/lox}$ control mice (and even their $Casp8^{lox/lox}Ripk3^{-/-}$ counterparts) in glucose clearance challenges, they did still exhibit dysregulated glucose tolerance, compared to ND-fed mice, with prolonged feeding (**Revised Fig 2D and E and Fig S5A-D**).

Therefore, at the later time point where insulin tolerance was tested for, it is not surprising that HFD-fed *Casp8^{LysMcre}Ripk3^{-/-}* mice also present with similarly impaired insulin responses, akin to control mice. It is possible that subtle differences in insulin responses may have been observable had the insulin tolerance test been performed earlier during the feeding regimen, but unfortunately, we are unable to retroactively collect this information.

R#3. In addition, in supplementary figure 2, the ITT is expressed as % relative to initial glucose. This way of quantification could lead to misinterpretation. I suggest using AoC (Area over the curve) suggested by Virtue and Vidal-Puig, Nat Met 2021 to be certain that there are no differences between the groups.

Response 21. We have now presented all ITT adjusted for starting starved baseline glucose levels as suggested by Virtue and Vidal-Puig, Nat Met 2021 (PMID: 34903886); please note that this does not change the overall pattern observed.

Fig S4D

Fig 3I

Fig S6H

Fig 3 and Fig S4D and S6H (Revised).

R#3. It is surprising that the conclusion is that MLKL does not contribute to inflammation given that there is a substantial decrease in the proportion of monocyte/macrophages in the liver (Figure 5). Can the authors clarify or revise this conclusion?

Response 22. We chose to reflect our conclusion based on other inflammatory parameters but have now performed additional analyses (NEW DATA Fig 4H) and indicated in the discussion that there were some inflammatory changes in the liver that may be related to MLKL expression.

R#3. The authors conclude that (line 255) "...reduced adipocyte hypertrophy and liver damage observed in aged or HFD-fed MLKL-deficient mice, suggests that MLKL activity alters tissue homeostasis to cause obesity-induced metabolic disease...." This conclusion arises from experiments in which they transplant either WT or MLKL KO BM into WT recipients. This conclusion seems to be too broad and not reflecting a focused analysis of their data. In reality, their data shows that MLKL (not sure you can say anything about its "activity") in non-haematopoietic cells alters tissue homeostasis to cause obesity-induced metabolic disease.

Response 23. We have amended the text to state that our data shows that MLKL expression in non-haematopoietic cells alters tissue homeostasis to cause obesity-induced metabolic disease.

October 17, 2024

RE: Life Science Alliance Manuscript #LSA-2023-02446-TR

Dr. Kate E Lawlor
Hudson Institute of Medical Research
Centre for Innate Immunity and Infectious Diseases
27-31 Wright Street
Clayton, Vic 3165
Australia

Dear Dr. Lawlor,

Thank you for submitting your revised manuscript entitled "Divergent roles for RIPK3 and MLKL in high-fat diet induced obesity and NAFLD in mice". We would be happy to publish your paper in Life Science Alliance pending final revisions necessary to meet our formatting guidelines.

- please be sure that the authorship listing and order is correct
- please add the Twitter handle of your host institute/organization as well as your own or/and one of the authors in our system
- please upload your table file as an editable doc or excel file

A. FINAL FILES:

B. MANUSCRIPT ORGANIZATION AND FORMATTING:

Sincerely,

Reviewer #1 (Comments to the Authors (Required)):

The authors have answered many of my concerns and significantly improved the manuscript. I have no further comments. I recommend publication

Reviewer #2 (Comments to the Authors (Required)):

The authors have responded to the reviewers' comments, and the manuscript is now suitable for publication in Life Science Alliance. However, I would like to request one additional change. In accordance with the current consensus on terminology, please update "NAFLD" and "NASH" to "MAFLD" and "MASH," respectively, throughout the manuscript.

October 30, 2024

RE: Life Science Alliance Manuscript #LSA-2023-02446-TRR

Dr. Kate E Lawlor
Hudson Institute of Medical Research
Centre for Innate Immunity and Infectious Diseases
27-31 Wright Street
Clayton, Vic 3165
Australia

Dear Dr. Lawlor,

Thank you for submitting your Research Article entitled "Divergent roles for RIPK3 and MLKL in high-fat diet induced obesity and MAFLD in mice". It is a pleasure to let you know that your manuscript is now accepted for publication in Life Science Alliance. Congratulations on this interesting work.

DISTRIBUTION OF MATERIALS:

Again, congratulations on a very nice paper. I hope you found the review process to be constructive and are pleased with how the manuscript was handled editorially. We look forward to future exciting submissions from your lab.

Sincerely,
